# The role of water in fault lubrication

Yijue Diao [1] & Rosa M. Espinosa-Marzal [1]

The friction between two adjacent tectonic plates under shear loading may dictate seismic activities. To advance the understanding of mechanisms underlying fault strength, we investigate the frictional characteristics of calcite in an aqueous environment. By conducting single-asperity friction experiments using an atomic force microscope, here we show three pathways of energy dissipation with increasing contact stresses: viscous shear of a lubricious solution film at low normal stresses; shear-promoted thermally activated slip, similar to dry friction but influenced by the hydrated ions localized at the interface; and pressure-solution facilitated slip at sufficiently high stresses and slow sliding velocities, which leads to a prominent decrease in friction. It is also shown that the composition of the aqueous solution affects the frictional response. We use this nanoscale evidence to scrutinize the role of brines on fault behavior and argue that pressure solution provides a weakening mechanism of the fault strength at the level of single-asperity contacts.

[1] Civil and Environmental Engineering, University of Illinois at Urbana-Champaign, 205 N. Matthews Avenue, Urbana, IL 61801, USA. Correspondence and requests for materials should be addressed to R.M.E-M. (email: rosae@illinois.edu)

When the static friction between two adjacent tectonic plates under shear loading is overcome, the two plates move relative to each other, which is known as a fault slip; under conditions of unstable slip, that is, when an intermittent motion (or stick–slip) occurs, an earthquake rupture may take place[1,2]. While earthquakes are one of the most notorious natural disasters, it remains elusive to accurately predict their occurrence and magnitude, which emphasizes the lack of understanding of the underlying mechanisms[3]. Along these lines, it has been proposed that water may play an important role in fault dynamics and earthquake nucleation[4], which has motivated this work.

Rate and state friction (RSF) constitutive equations[3,5] are commonly used to phenomenologically describe the frictional response of tectonic fault zones and earthquake nucleation. According to RSF equations, only velocity-weakening friction, i.e., a decreasing friction with an increase in slip rate, can yield an unstable fault slip. Regarding the effect of water, RSF equations predict that fluid overpressure decreases the critical fault stiffness, and therefore, it should promote aseismic creep. In contrast to these predictions, seismologic studies[4] as well as laboratory experiments[6] suggest that a high fluid pore pressure can trigger a dynamic slip instability, indicating that fault weakening—likely as a result of the reduced effective (compressive) stress, which brings the fault closer to failure—can overcome velocity-strengthening friction. Other mechanisms have been proposed to explain the effect of water on fault friction. For instance, water may promote fracture of asperities by chemically reacting with the strained apexes of microcracks (also called subcritical crack growth)[7–9], thereby lowering the resistance to slip, i.e., friction. The weakening of wet fault gouges may also rely on intergranular lubrication facilitated by water adsorption to the grain surface[10,11], like the oil lubricating an engine. Velocity-weakening friction and stick–slip were observed in experiments on water-saturated limestone gouge at high temperature (150 °C), whereas stable sliding was observed under dry and low temperature conditions[10]. Here, the authors could not unambiguously identify the underlying mechanisms, and proposed that crystal plasticity at asperity contacts coupled to granular flow and perhaps pressure solution—a process by which the contact stress leads to mineral dissolution, coupled to diffusion of the solutes and re-precipitation of the mineral on unstressed regions[12,13]—could play a role[14]. In fact, pressure solution has been concurrently observed in other experiments with water-saturated calcite-rich gouge that showed a reduction in friction compared to dry conditions[10]. Controversially, re-precipitation of calcite after its pressure-induced dissolution has been also related to overall gouge compaction[15], and thereby, to an increase in the friction coefficient[16]. The complexity of the investigated systems has hindered discerning the contribution of these multiple mechanisms to the frictional response of fault materials in the presence of water, which calls for fundamental research on more simple systems.

Atomic force microscopy (AFM) is a powerful tool to investigate the fundamental mechanisms of frictional behavior at the nanoscale[17,18]. Although AFM experiments are highly idealized compared to the complexity in nature and in laboratory experiments with simulated fault gouge, they allow isolating and inspecting physical and chemical mechanisms. For instance, a previous AFM study demonstrated that the origin for the increase in the static friction of single-asperity silica–silica contacts of nanometer size is the formation of chemical bonds between the silica surfaces[18–20]. A precedent study of the frictional characteristics of calcite under dry conditions by AFM showed that both stick–slip behavior and kinetic friction depend on the crystallographic sliding direction[21]. While these recent works

illustrate the first attempts to bridge the gap between nanoscale friction measurements and fault dynamics, many questions still remain open.

Recognizing that carbonate-dominated lithologies play an important role in mid- to shallow-crustal faulting[16], the purpose of this article is thus to rationalize the frictional characteristics of calcite in the presence of an aqueous solution at nanoscale. An AFM was used to measure the friction force as a function of the applied load and sliding velocity. By sliding an oxidized silicon tip along the atomically flat calcite $(10\bar{1}4)$ plane, the friction force was measured while isolated from other processes occurring concurrently in natural systems and macroscale experiments. An AFM tip with a silica surface was selected because its frictional characteristics are similar to those of silicate rocks[18,22]. The selected range of normal stresses is relevant to that along faults in the upper Earth crust, which is in the order of 100 MPa[23], but higher in the multiple-asperity contacts that determine the fault strength. Scrutinizing the change in friction caused by varying stress, velocity, and brine concentration helps to understand the pathways of energy dissipation, i.e., the origin of friction. Based on the measured frictional behavior of calcite in aqueous solutions, three different mechanisms of energy dissipation are distinguished (Fig. 1): viscous dissipation at low stresses (abbreviated as LS, Hertzian contact stresses $\sigma_n \leq 200$ MPa), shear-promoted thermally activated slip at intermediate stresses (IS, $200 < \sigma_n \leq 400$ MPa); and pressure-solution facilitated slip at higher stresses (HS, $\sigma_n > 400$ MPa). This work thus provides evidence for the presence of a lubricating film composed of ions and water that remains confined between calcite and silica surfaces even under high applied pressures. Our experimental study also demonstrates that pressure-induced dissolution of calcite at sufficiently high stresses and slow sliding velocities can cause a significant reduction of friction and facilitate slip, which is proposed to be a potential mechanism of fault-strength weakening at the level of single-asperity contacts.

## Results
**Viscous shear and hydration lubrication in the LS regime.** Figure 1a–e show the friction force as a function of velocity at applied loads of 0.5, 1, and 2 nN ($\sigma_n \leq 200$ MPa, Supplementary Table 1) and at calcium chloride concentrations of 0 mM (no $CaCl_2$ added), 1, 10, 100 mM, and 1 M to simulate calcium concentrations in nature, from ground water[24] to concentrated brines[25]. As shown here, friction increases with load but the change is very small in this narrow range of loads (0.5–2 nN). The velocity dependence of the friction force in the LS regime can be fit by a power law at all selected concentrations (see solid lines), which is reminiscent of the shear response of fluids as described by the Ostwald–de Waele equation[26]

$$F_L \sim A\eta_0(v/D)^n, \tag{1}$$

where $F_L$ is the friction force, $A$ is the area, $\eta_0$ is a constant, $v/D$ is the shear strain rate, $D$ is the fluid film thickness, and $n$ an exponent. This relation suggests that friction in the LS regime is mainly due to a viscous force that results from shearing the confined aqueous solution. The lubrication mechanism provided by a nanoconfined solution has been previously referred to as "hydration lubrication"[27].

The composition of the confined aqueous solution between calcite and a (micron-sized) silica tip was scrutinized in our previous AFM study[28], by measuring the normal surface forces down to subnanometer calcite–silica separations. This work showed that a strong repulsion between the confining surfaces— also termed disjoining pressure—is originated by the colloidal forces between the two confining surfaces across the thin film of

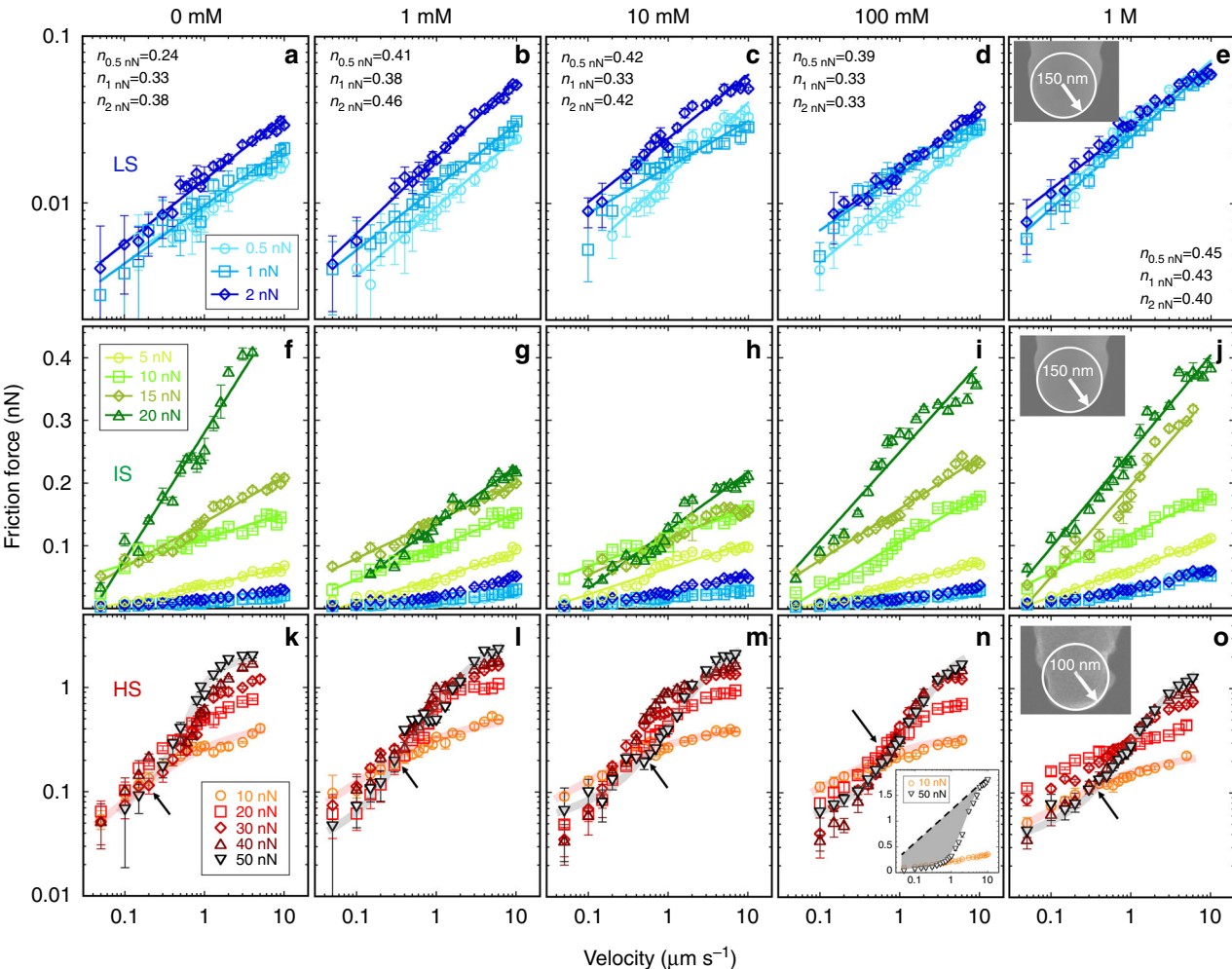

**Fig. 1** Friction between calcite and an AFM tip as a function of velocity in $CaCl_2$ solutions equilibrated with calcite. $CaCl_2$ concentrations are selected as **a**, **f**, **k** 0 mM, **b**, **g**, **l** 1 mM, **c**, **h**, **m** 10 mM, **d**, **i**, **n** 100 mM, and **e**, **j**, **o** 1 M. The friction force in **a–e** LS regime (0.5–2 nN, $\sigma_n \leq 200$ MPa) and **k–o** HS regime (20–50 nN, $\sigma_n > 400$ MPa) is plotted as a function of speed in log–log scale, while it is plotted in linear-log scale in **f–j** IS regime ($\sigma_n \sim$ 200–400 MPa). Friction increases with velocity under all conditions. The calculated contact stresses at each load are summarized in Supplementary Table 1. The solid lines show the fits of **a–e** Eq. (1) and **f–j** Eq. (2) to the experimental data. The exponent $n$ is shown for each load in **a–e**. Error bars give the variation in friction across ten friction loops; the variation is often so small that the error bars are smaller than the symbol size, and thus, difficult to see. The blue shades are for LS regime, green for IS regime, and red for HS regime, while the intensity (from light to dark) indicates an increase of load (for each regime). Inset of **e** and **j** shows a scanning electron microscopy image of the tip used in LS and IS regimes (with a radius of ~150 nm), and the inset of **o** shows the tip (with a radius of 100 nm) used in the HS regime. Inset of **n** shows the velocity-dependent friction force in linear-log scale at two selected loads, 10 and 50 nN, and the shadow qualitatively indicates the reduction of the frictional strength mediated by pressure solution. The arrows in **k–o** point at the crossover of the velocity-dependent friction force and is the qualitative footprint for pressure solution of calcite

solution, and it prevents the solution to be squeezed-out under an applied pressure[29–31]. The confined solution has a thickness of a few nanometers and consists of water layers that are strongly adsorbed to calcite and located underneath a layer of partially dehydrated calcium ions, and fully hydrated calcium ions further away from the surface.

Assuming that the change in $D$ is negligible across the investigated range of loads and velocities, the average exponent $n$ for all selected concentrations is estimated to be $0.38 \pm 0.06$. An exponent $n$ smaller than 1 indicates that the confined fluid behaves as a non-Newtonian fluid with shear-thinning behavior, i.e., its viscosity decreases with increasing shear strain rates. A calculation of the viscosity is, however, not possible, because the absolute separation between the tip and the calcite surface ($D$) cannot be unambiguously determined. Nevertheless, a similar shear-thinning behavior has been observed for confined water between mica and an AFM tip at stresses $\geq 100$ MPa[32], while a

constant viscosity was reported at much smaller normal stresses ($< 1$ MPa)[27]. This suggests that the high normal stresses applied in the LS regime may be responsible for the observed non-Newtonian behavior of the confined aqueous solution.

**Shear-promoted thermally activated slip in the IS regime.** Figure 1f–j show the friction force as a function of velocity in the IS regime ($\sigma_n \sim$ 200–400 MPa). The friction force scales with the logarithm of velocity, i.e., $F_L \sim \ln(v)$, thereby indicating a mechanism of energy dissipation that differs from the viscous dissipation in the LS regime. This logarithmic relation is attributed to a shear-promoted thermally activated slip in the context of transition state theory[33]. This theory was applied by Eyring[34] to describe liquid viscosity at the molecular level based on activated flow and is often applied to describe the origin of friction in thin-film lubrication[35]. That is, for slip to occur, the molecule,

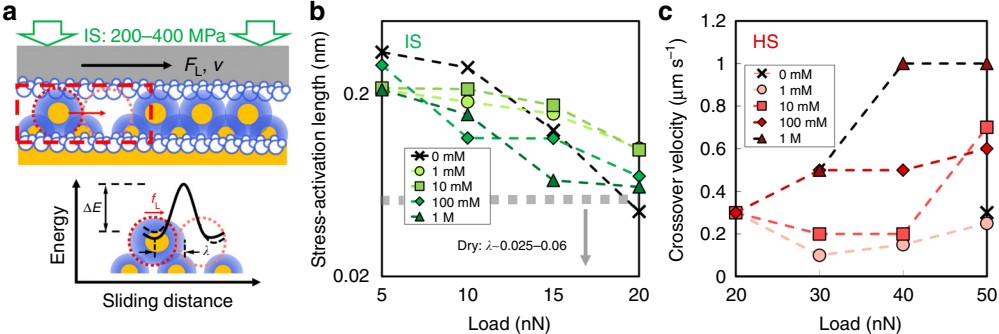

**Fig. 2** Stress-activation length and crossover velocity. **a** Schematics of the shear-promoted thermally activated slip in the IS regime ($\sigma_n \sim 200$–400 MPa). **b** Stress-activation length $\lambda$ obtained from fitting Eq.(2) to the velocity-dependent friction force in the IS regime. The error bars show the root-mean-square error of the fits. The region under the gray dashed line shows the range of $\lambda$ in dry experiments, which is described here by the shear-promoted thermally activated slip of the surface atoms for comparison purposes. Tip radius = 150 nm. **c** Crossover velocities $v_c$ in the HS regime ($\sigma_n > 400$ MPa); tip radius = 100 nm. The results in **b**, **c** are shown for the selected CaCl$_2$ concentrations displayed in the legends

initially in an equilibrium position (an energy minimum) needs to jump over a "transition state" before reaching the adjacent energetic minimum, which requires an activation energy ($\Delta E$) (Fig. 2a). Although the thermal energy of the molecules might be sufficient to overcome this energy barrier, the shear force applied on the molecule reduces the energy barrier, and thereby, promotes the slip; when the molecule falls in the adjacent minimum, the applied work is irreversibly dissipated (lost). Although described at the molecular level, this theory has been often applied to explain friction in macroscale contacts, as well[36,37].

Based on our previous work[28], we assume that hydrated calcium ions are not squeezed-out under high applied pressures, but instead they remain localized on oppositely charged surface sites and undergo similar thermally activated slip, as shown in Fig. 2a. Considering that the jump frequency of the hydrated ions $f$ is increased by the applied lateral force according to $f \sim f^* \exp(-(\Delta E - F_L \Omega / A)/k_B T)$, $F_L \Omega / A$ involving the decrease of the energy barrier imposed by the lateral force, $\Omega$ the stress activated volume, and $f^*$ the characteristic frequency of water and ions, the following expression[27] is obtained for the friction force $F_L$:

$$F_L = \frac{\Delta E}{\lambda} + \frac{k_B T}{\lambda} \ln(v/v_0), \qquad (2)$$

where $k_B$ being the Boltzmann constant, $T$ is the absolute temperature, $v = d \cdot f$ is the sliding velocity, $d$ being the jump amplitude, $v_0 = d \cdot f^*$ is the reference velocity, and $\lambda = \Omega / A$ is the stress-activation length. The solid lines in Fig. 1f–j show the fits of Eq. (2) to the experimental results. $\lambda$ represents the characteristic length over which the confined ions and water can build up strain in response to shear, and it appears in Fig. 2a as the distance from the initial equilibrium position to the transition state. The calculated $\lambda$ significantly decreases with increasing normal loads at all investigated concentrations (Fig. 2b); that is, the localized ions and water can build up strain over shorter distances, which suggests a decrease in mobility of the confined solution with increase in normal stress.

As shown in Fig. 1f–j, friction collectively decreases when the salt concentration increases from 0 to 10 mM, while it increases by further increasing the concentration to 1 M, thereby indicating the action of at least two phenomena with opposite effects. Furthermore, the non-monotonic change in friction with concentration coincides with the inverse trend of $\lambda$, first increasing and then decreasing with rising concentration. To better understand these results, reference measurements were made with mica in NaCl solutions (Supplementary Fig. 3). CaCl$_2$ was not selected here because it had been previously shown that

although calcium ions adsorb on mica[38], the hydration force between mica surfaces in CaCl$_2$ solutions does not reveal the presence of multiple layers of ions and water at concentrations below 100 mM[30,39,40]. In contrast, hydrated sodium ions accumulate on mica surfaces and form multiple layers at lower concentrations, similar to our observations for calcium ions on calcite[28]. Interestingly, we also observed opposite trends of friction and $\lambda$ on mica in NaCl solutions. Supplementary Fig. 3 shows a decrease of friction and a concurrent elongation of $\lambda$ by increasing the concentration from 0 to 1 M. Previous studies have proved that a higher amount of strongly hydrated sodium ions accumulate at the mica surface with increasing concentration[41]. Further, theory has showed that the adsorption of ions to surfaces leads to an electric field in the neighboring water molecules[42], thereby creating boundary films with high fluidity[43], while the fluidity of the hydration shell of the ions has been proposed to facilitate slip[27]. Accordingly, the observed increase of $\lambda$ with concentration in Supplementary Fig. 3 reflects the enhanced fluidity or mobility of the confined solution, since more water is retained in the contact within the hydration shell of the adsorbed sodium ions, which decreases the resistance to sliding, i.e., friction.

In the experiments on calcite, the non-monotonic trend of friction (and of $\lambda$) with increasing concentration is intriguing, since our previous work also revealed an increasing amount of calcium ions populating the calcite–solution interface with increasing concentration[28]. This suggests that another mechanism is at play. Indeed, the increasing friction force at concentrations above 10 mM coincides with the charge reversal of the confined calcite surface[28]. We thus speculate that charge reversal of calcite (from negative to positive) magnifies the energy barrier at concentrations above 10 mM due to an additional electrostatic attraction to the naturally oxidized silicon tip, which is negatively charged. As a result, the increase in friction force at concentrations above 10 mM can be justified. Although the conditions for charge reversal are mineral-specific, this phenomenon is expected to be universal for mineral surfaces in contact with calcium-rich brines, and thus, to affect friction.

It is important to discuss the influence of the AFM tip on these results. First, analogous results were obtained with a silicon nitride AFM tip (Supplementary Fig. 4a), and hence, the bulk material of the tip does not affect the mechanism of energy dissipation. Second, the friction force measured with a naturally oxidized silicon tip sliding on a silica surface decreases with velocity (Supplementary Fig. 5a), which has been attributed to the formation of hydrogen and/or siloxane bonds between the silica surfaces[19,44]. Therefore, the identified shear-promoted thermally

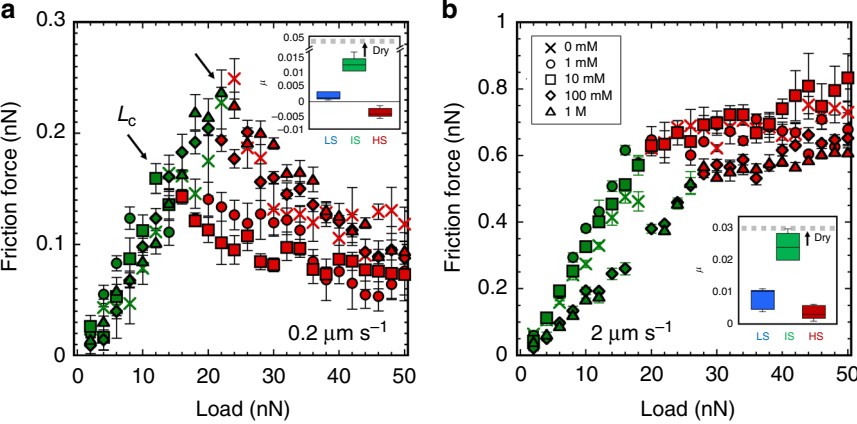

**Fig. 3** Friction between calcite and an AFM tip as a function of load in $CaCl_2$ solutions. The selected concentrations are shown in the legend in **b**. The results are shown at sliding speeds of **a** 0.2 µm s$^{-1}$ and **b** 2 µm s$^{-1}$, in green for the IS regime and in red for the HS regime. Error bars show the variation in friction over ten friction loops. Tip radius = 100 nm. The insets show the friction coefficient ($\mu$)—calculated as the slope of the friction force vs. normal load—in the LS, IS, and HS regimes at the two selected sliding velocities; the friction coefficient under dry conditions is shown as reference (dashed gray line). The box diagrams illustrate the variation of $\mu$ with $CaCl_2$ concentration; the error bars give the standard deviation of $\mu$. The friction coefficient in the LS regime was calculated with experimental data measured with an AFM tip with a smaller stiffness (not shown here). The friction coefficient in the HS regime was calculated above the load $L_c$ and does not include the plateau at the highest loads for simplification. The dependence of friction on velocity shown in Fig. 1 is reflected in the increase of the friction coefficient with velocity (0.2 and 2 µm s$^{-1}$); note the different scale on the Y-axis

activated slip in Fig. 1f–j mainly reflects the frictional behavior of calcite.

**Pressure-solution facilitated slip in the HS regime**. Figure 1k–o show the friction force as a function of the velocity in the HS regime ($\sigma_n > 400$ MPa). Owing to the smaller radius of the tip used in these experiments (100 nm, instead of 150 nm in IS regime), the contact stress is higher than in the IS regime ($\sigma_n \sim$ 200–400 MPa) at the same applied load. Remarkably, a decreasing friction with an increase in normal load is observed at slow sliding velocities under several conditions (e.g., see Fig. 1m), which was confirmed in load-dependent friction measurements at selected sliding velocities (Fig. 3). Figure 3a shows that friction decreases with increasing load in the HS regime; note the change from IS regime (green) to HS regime (red). The decrease in the friction coefficient $\mu$—i.e., the slope of $F_L$ vs. $L$—is more significant at slower sliding velocities (compare Fig. 3a, b). Importantly, when analogous friction measurements are conducted in a dry nitrogen atmosphere, i.e., in the absence of the aqueous solution, the friction force increases monotonically with applied load at all sliding velocities (Fig. 4a), which emphasizes the key role of the solution in facilitating slip. The insets in Fig. 3a, b display the calculated friction coefficients for these measurements to emphasize the remarkable drop in friction in the HS regime. Although $\mu$ appears negative in Fig. 3a, it should be noted that a quasi-plateau is achieved at the highest loads, but this transition is not quantified, for simplification. Such a decrease of the friction coefficient was not detected for mica in aqueous solution, where mica is effectively insoluble (Supplementary Fig. 6). Instead, friction increased linearly with load first, until it abruptly jumped up. A comparison from thermodynamics perspective reveals a significant difference between calcite and mica. The hydration energy of calcite is reported to be 94 kJ mol$^{-1}$ [45], which is higher than the maximum applied work (~27 kJ mol$^{-1}$, see calculations in the caption of Supplementary Fig. 6). This supports that calcite remains hydrated at the maximum applied pressure. In contrast, a similar estimation for mica shows that surface dehydration takes place at an applied load of ~60 nN, and hence, the observed increase in friction in Supplementary Fig. 6 is likely originated by

the dehydration of mica, thereby resulting in a dry contact. This comparison emphasizes the key role of both the strong affinity of the aqueous solution to calcite and its solubility in decreasing friction at high stresses.

At the sliding conditions at which the friction coefficient decreased, AFM imaging of the calcite surface showed the track of the tip (Supplementary Fig. 7), while re-precipitation sometimes occurred close to the track (see the bright spot in Supplementary Fig. 7c). Images taken after similar friction measurements in ethanol, where calcite is insoluble, do not show any change of the surface, which excludes brittle fracture and plastic deformation of calcite (Supplementary Fig. 8). The tracks are similar to the previously reported wear of a single-crystal calcite created by an AFM tip along the edges of pre-existing etch pits in aqueous solution[46]. Hence, the remarkable drop in friction with increase in load is attributed to the activation of the pressure solution of calcite at sufficiently high normal stresses and slow sliding velocities. As described earlier, the dehydration of calcite cannot happen at the highest applied pressure in this work, and hence, strongly adsorbed water and ions are still retained between the two confining surfaces, thereby providing a path for the pressure-induced dissolution of calcite. We hypothesize that if the calcite under the AFM tip could be dissolved and re-hydrated before the tip slides away, it would facilitate slip in a similar fashion as in the LS regime, that is, by hydration lubrication, but at much higher contact stresses.

The decrease of the friction coefficient shown in Fig. 3 is reflected in Fig. 1k–o as a crossover of the velocity-dependent friction force when the load is decreased (see arrows). Such a crossover was not observed in reference tests in the absence of the aqueous solution, as illustrated in Fig. 4a. Therefore, the crossover is considered to be a qualitative footprint for pressure-induced dissolution of calcite. Although one cannot rule out the possibility that pressure solution also takes place in the IS regime, both the relation $F_L \sim \ln(v)$ and the increasing $F_L$ with load in this regime indicate that the influence of pressure solution, if any, is small. By the same token, pressure solution, if any, is small at 10 nN in Fig. 1k–o, and hence, this load is taken as reference to quantify the crossover velocity, which is shown in Fig. 2c. The general trend that higher loads lead to higher crossover velocities is

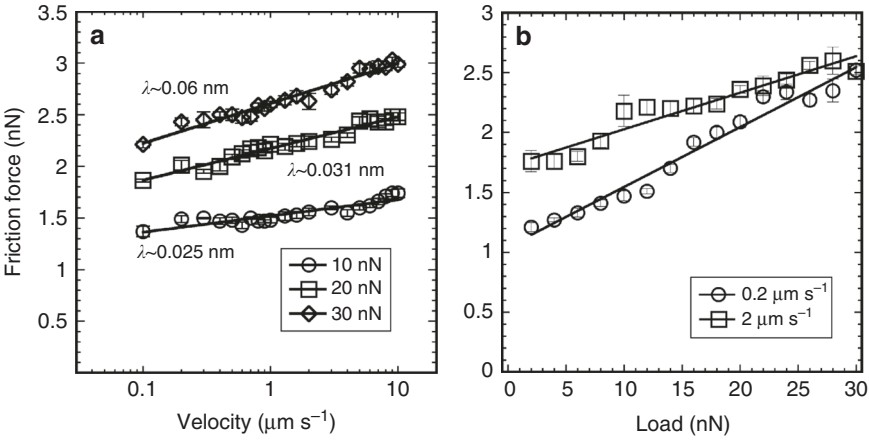

**Fig. 4** Friction between calcite and an AFM tip in a dry nitrogen atmosphere. Friction force as a function of **a** velocity and **b** applied load. Tip radius = 100 nm. Under dry conditions, the friction force—higher than in wet experiments—still scales with the logarithm of the sliding velocity at contact stresses as high as 620 MPa. For comparison purposes, friction is described as a shear-promoted thermally activated slip that accounts for the rupture of the adhesive bonds between the surface atoms on calcite and tip[17]. Stress-activation lengths $\lambda$ obtained by fitting Eq. (2) to the velocity-dependent friction force are shown in **a** for each load

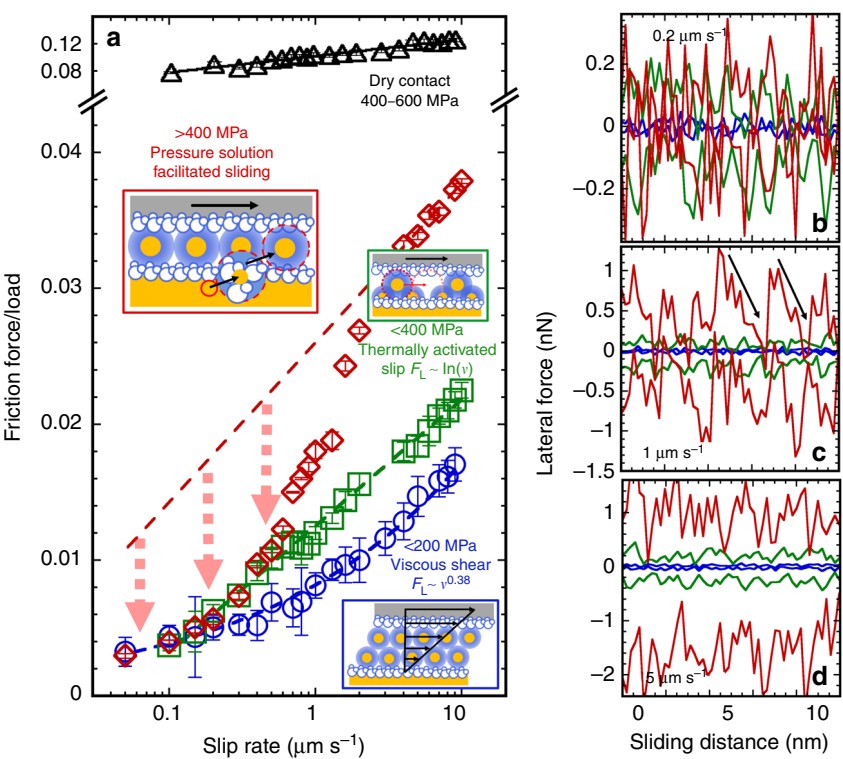

**Fig. 5** Velocity-dependent frictional strength in LS, IS and HS regimes and corresponding lateral forces as a function of the sliding distance.
**a** Rate-strengthening frictional strength (shown as the friction force divided by the applied load) of a single-asperity contact between calcite and an AFM tip in a 100 mM $CaCl_2$ solution at low ($\leq 200$ MPa, blue circles, LS), intermediate (200–400 MPa, green squares, IS) and high normal stresses ($> 400$ MPa, red diamonds, HS), with the schematics showing the three identified friction mechanisms. The black symbols represent the frictional strength of the dry single-asperity contact (Fig. 4) for comparison. At low contact stresses (LS regime), the brine lubricates the single-asperity contact causing a significant decrease in friction when compared to the dry contact (black). By further increasing the stress (200–400 MPa, IS regime); the response to shear in this regime is described as a shear-promoted thermally activated slip of the localized hydrated ions on the surface, which leads to a logarithmic dependence of friction on velocity over the investigated range ($\leq 10 \, \mu m \, s^{-1}$). At high contact stresses (> 400 MPa, HS regime), the pronounced weakening of the single-asperity contact results from pressure-induced dissolution of calcite, if the contact time is sufficiently large. The red dashed line gives the calculated friction force according to Eq. (2), as in the IS regime, and extrapolated to lower velocities; the arrows illustrate the decrease in the frictional strength provided by pressure solution. **b–d** Representative lateral force as a function of the sliding distance in the three regimes at selected slip rates. The color scheme is the same as in **a**, i.e., blue for LS regime, green for IS regime, and red for HS regime. Note the different scale on the $Y$-axis

consistent with the proposed mechanism: pressure solution is enhanced with increasing normal stresses, requiring less time for dissolution, and therefore, taking place at higher sliding velocities compared to lower stresses. Furthermore, the increasing crossover velocity with increasing concentration indicates the more prominent pressure solution under these conditions. This is most likely originated by the increasing reactivity of calcite with increase in calcium concentration (and the concurrent decrease in pH, Supplementary Table 1), as inferred from the more dramatic reconstruction of the unconfined calcite surface (Supplementary Fig. 9).

The proposed weakening mechanism is also consistent with the estimated kinetics of pressure-induced calcite dissolution. Taking a normal stress $\sigma_n \sim 540$ MPa, the chemical potential $\Delta\mu$ that drives the dissolution of the stressed calcite is given by $\Delta\mu/k_B T = V_m\sigma_n/k_B T = \ln(a^p/a^0) \sim 7.87$, $V_m$ being the molecular volume of calcite, and $a^0$ and $a^p$ the activity of the solution in equilibrium with stress-free and stressed calcite, respectively[47]. The increase in chemical potential induced by $\sigma_n$ leads to a sudden increase of the local solubility as well as a dramatic undersaturation of the interfacial solution. The reaction-limited dissolution rate under such conditions is proportional to the solution activity[48]. Thus, the pressure-induced dissolution rate is estimated as $r^p \sim r^0 a^p/a^0$, being $r^0$ the dissolution rate of stress-free calcite. Let us now take $r^0 \sim 1.8 \times 10^{-10}$ mol cm$^2$ s$^{-1}$ as the reaction-limited dissolution rate of calcite in water[49], equivalent to ~1 formula unit (CaCO$_3$) nm$^{-2}$ s$^{-1}$. At the assumed contact stress, the rate will increase to $r^p \sim e^{7.87}$ formula units nm$^{-2}$ s$^{-1}$. Given the unit cell of calcite $c \sim 0.5$ nm, the time required for a formula unit to be dissolved is $\Delta t = 1/(c^2 r^p) \sim 1.5$ ms. Taking $d = 6.8$ nm as the estimated diameter of the Hertzian contact area, it requires a velocity slower than $v_c = d/\Delta t = 4.43$ µm s$^{-1}$ for a unit cell to be pressed by the AFM tip longer than $\Delta t$. Theoretically, sliding events at either slower velocities (to allow longer tip–substrate contact time), or higher stresses (to enhance $r^p$ further) should let calcite dissolution happen while the tip slides. In the range of stresses in the HS regime (~400–700 MPa) the calculated $v_c$ lies between 0.7 and 94 µm s$^{-1}$, which agrees well with the range of velocities in our study. We thus conclude that pressure solution of calcite may cause a decrease in friction by promoting hydration lubrication at high normal stresses and sufficiently slow sliding velocities.

In summary, we have resolved three different mechanisms responsible for the frictional strength of this nanoscale asperity. Figure 5a shows the friction force divided by load as a function of the slip rate in the three regimes; note that in the HS regime this value significantly differs from the friction coefficient, which becomes negative under certain conditions (Fig. 3a). Dry friction is also displayed for comparison. Representative measurements for the lateral force as a function of the sliding distance (during trace and retrace of the tip) at three selected sliding velocities are shown in Fig. 5b–d. Hydration lubrication occurs at normal stresses $\sigma_n \leq 200$ MPa, when the confined solution lubricates the nanoscale contact and friction is small and dictated by the viscous shear force of the fluid film. Here, the corrugation of the lateral force (Fig. 5b–d, blue) is indistinguishable from the noise (~0.02 nN), supporting smooth sliding, and thus, viscous energy dissipation. At higher normal stresses ($\sigma_n \sim 200$–400 MPa, green), velocity-strengthening friction is originated by the shear-promoted thermally activated slip of the surface-localized hydrated ions, and it is strongly dependent on the solution composition. Here, the variation of the lateral force is much higher than the noise, revealing a periodic atomic-scale stick–slip with a slip length of ~0.5 nm, close to the unit cell of calcite, throughout all sliding velocities. Such discontinuous sliding is characteristic of the shear-promoted thermally activated slip in

nanoscale experiments as a result of the jumps from the transition state to the adjacent energy minimum[17,50]. Intermittent slip prevails in the HS regime (red), although it becomes more irregular, which is likely associated with the concurrent pressure solution of calcite. The pressure-induced dissolution re-activates hydration lubrication at higher stresses ($\sigma_n > 400$ MPa), thereby causing a significant weakening of the frictional strength of this single asperity. Here, the red dashed line shows the calculated friction force assuming that the friction force at the highest velocities is still given by Eq. (2) and extrapolating to lower velocities; the area between calculated and measured friction force gives the decrease in dissipated energy per unit time as a result of pressure solution (see arrows in Fig. 5a). This is most prominent at slow sliding velocities (0.2 µm s$^{-1}$, Fig. 5b), where the lateral force in the HS regime is of similar magnitude as in the LS and IS regimes but at much higher normal stresses. The lateral force increases when sliding occurs at 1 µm s$^{-1}$, while the reproducible drops pointed by the arrows in Fig. 5c reflect the sequential weakening of the frictional strength upon sliding. The characteristic drops vanish with a further increase in the sliding velocity to 5 µm s$^{-1}$ (Fig. 5d), indicating that pressure-solution facilitated slip is not noticeably active anymore. This mechanism is more pronounced with increasing calcium concentration, likely owing to the enhanced calcite surface reactivity; in nature, it will depend on the specific composition of the brine.

## Discussion

Implications of our work regarding the frictional strength of calcite-rich faults are discussed next. Our experiments have shown that the presence of the aqueous solution leads to a significant weakening of the frictional strength of the single-asperity contact compared to dry conditions, as the friction coefficient drops from ~0.05 to lower than ~0.005 at slow sliding velocities (Fig. 3a). Previous studies of calcite-rich fault gouge also reported a decrease in the friction coefficient from ~0.7 to ~0.6 under saturated conditions; however, much less pronounced[10]. We also note the significantly lower friction coefficients reported in our work compared to macroscale friction coefficients in fault gouge and rocks. There are multiple reasons for such discrepancy, as demonstrated previously for other substrates like mica[27,51], quartz[43,52], and graphite[53]. In the case of fault gouge and rock friction, multi-asperity interlocking during macroscopic sliding of such rough surfaces as well as asperity fracture can cause a significant increase in friction that is excluded in atomically smooth contacts, like in our nanoscale experiments. The simple geometry of our study enables us to evaluate the direct effect of hydration lubrication and pressure solution on the frictional characteristics of calcite at the level of a single asperity.

The strong affinity of the aqueous solution to the calcite surface results in the presence of a thin fluid film between the confining surfaces at high normal stresses. The confined fluid film is thus responsible for hydration lubrication under shear loading, serving as an excellent lubricant. In the case of the calcite–silica interface, a high disjoining pressure was measured (up to ~200 MPa), which makes hydration lubrication effective in this range of normal stresses[28]. While higher stresses bring the friction coefficient closer to that of the dry contact (insets of Fig. 3), we propose that hydration lubrication may be also active at much higher stresses (> 400 MPa), if pressure-induced dissolution happens concurrently with the slip. In fact, the drop in the friction coefficient in the HS regime is remarkable at the slowest slip rates. We have argued why the slip rate is thus an important parameter to predict the onset of pressure-solution facilitated slip based on the kinetics of mineral dissolution. A comparison between the dissolution kinetics of the mineral and the contact time during slip, as

described in this work, should also define the conditions for pressure-solution facilitated slip in natural systems. In agreement with this, there is evidence for the dependence of the pressure solution on the slip rate in macroscopic experiments. For instance, grain size reduction due to fluid-assisted solution transfer was only observed at sufficiently slow sliding ($< 30 \, \mu m \, s^{-1}$) in calcite-rich gouge experiments[16]. While the role of ion diffusion and other phenomena, like electrochemical potential gradients[54], have been neglected so far, they may either retard or enhance pressure solution. This still requires more focused investigations.

The velocity-strengthening frictional behavior observed in this work under all conditions agrees well with previous studies on water-saturated calcite-rich gouges in the same range of sliding velocities and temperature[10,16,55]. It is to be noted that these studies also showed a transition into velocity-weakening frictional behavior[16,56,57], yet at sliding velocities greater than investigated in this work due to the limitations of our instrument, and hence, a discussion of this behavior is out of the scope of this work. RSF constitutive equations predict that a velocity-strengthening frictional response should favor aseismic creep[55], and therefore, calcite gouge would be expected to resist initial rupture nucleation at slow slip rates ($< 10 \, \mu m \, s^{-1}$) and over a wide range of contact stresses according to our nanoscale experiments. Nevertheless, prior work has shown that under stress/temperature conditions typical of the occurrence of fluid-induced seismicity, a wide variety of fault materials exhibits velocity-strengthening frictional behavior in the laboratory, which emphasizes the limitations of the standard RSF analysis. For instance, it has been recently reported that a high fluid pore pressure in carbonate-bearing fault gouge resulted in gouge dilation and was able to trigger fast acceleration and earthquake slip, despite its velocity-strengthening frictional behavior[6]. The effect of the disjoining pressure can be considered analogous to that of the pore fluid overpressure in decreasing the effective normal stress, and thereby in bringing faults closer to failure according to the Coulomb–Mohr relation. Moreover, according to our work, the nanoconfined fluid enables hydration lubrication, which appears as the origin of a dramatic weakening of the fault strength of single-asperity contacts; especially at high contact stresses when hydration lubrication is re-activated by pressure solution of calcite at the multi-asperity contacts. It should be noted that (long-term) slip weakening has been observed to overcome the velocity-strengthening behavior and trigger fault instability[58]. But even in the case of stable frictional behavior, fault creep—as a result of the significant decrease in frictional strength—will result in stress transfer in the surrounding, and hence, possibly, earthquake triggering on adjacent fault patches that could be more prone to develop frictional instabilities.

In conclusion, the simple geometry in our study and the selected mechanical and chemical conditions have enabled us to evaluate the direct effect of hydration lubrication and pressure solution on the frictional strength of calcite. We have demonstrated that pressure solution of calcite can directly impair the frictional strength of single-asperity contacts providing hydration lubrication, similar to that occurring at much lower contact stresses. This nanoscale evidence can be extended to fault weakening: when a fault undergoes creep slow enough for pressure solution to come into play, a slip can be triggered more favorably, which alters fault dynamics at the level of single-asperities. We thus hypothesize that earthquakes ruptures might be triggered and perhaps propagated by fault weakening induced by pressure solution of calcite asperity contacts. Our results have also shown that the influence of the brine composition on mineral reactivity should be considered. The relative contributions of subcritical crack growth, crystal plasticity, lubrication, and pressure solution to macroscopic friction still remain to be quantified. The range of applied pressures and slip rates investigated in this study is pertinent to earthquake nucleation, while the additional effect of Earth's crust temperature at greater depths on the proposed mechanisms calls for future studies.

## Methods

**Sample preparation**. To simulate calcium concentrations of water bodies in nature, from ground water[24] to concentrated brines[25], calcium chloride $CaCl_2$ (purity $> = 99.0\%$, Sigma-Aldrich) was dissolved at room temperature in ultrapure water (18.2 M$\Omega$ cm resistivity) to achieve concentrations of 0 mM (no $CaCl_2$ added), 1, 10, 100, and 1 M. All solutions were then equilibrated with excessive calcite ($CaCO_3$) powder overnight (purity $> = 99.0\%$, Sigma-Aldrich) and filtered prior to use. Ion concentrations of the $CaCl_2$ solutions and pH values can be found in Supplementary Table 1. Freshly cleaved Iceland Spar calcite crystals fixed to glass slides were immersed in the solutions with the ($10\bar{1}4$) plane exposed to solutions. The home-made fluid cell was covered by a membrane to minimize evaporation during experiments. The calcite surface was equilibrated for 1 h in each solution before performing the friction-force measurements.

**Friction force measurements**. A Nanowizard AFM (JPK Instruments, Germany) was used to measure the friction force; the mechanic model is shown in the Supplementary Fig. 1. All the experiments reported here were performed with the same AFM cantilever type (CSC38/no Al, Mikromasch) with normal spring constants ranging between 0.1 and 0.3 N m$^{-1}$, as determined by the thermal calibration method[59]. To measure friction, a naturally oxidized silicon tip was slid along the atomically flat calcite ($10\bar{1}4$) plane; the sliding direction was maintained constant during each experiment. The tip radius of the reported experiments ranges from 100 to 150 nm, as determined by scanning electron microscopy (SEM) imaging. Considering that the smallest reliable force that the instrument can apply is ~0.2 nN, such large tip radii were selected to allow applying pressures smaller than the solvation pressure[60]. Nevertheless, replica experiments were conducted with smaller tip sizes to confirm the reproducibility of results. SEM imaging of the AFM tips before and after friction-force measurements showed no significant changes in tip size, thereby excluding any irreversible change of the AFM tip, e.g., by wear or plastic deformation, during the duration of the experiment.

AFM imaging in contact mode was first conducted searching for an atomically smooth 100 nm × 100 nm surface area (RMS roughness < 100 pm, Supplementary Fig. 2). Isothermal friction-force measurements at 25 °C were made on an atomically flat calcite surface along the ($10\bar{1}4$) plane as a function of applied load ($L$) and velocity ($v$) at each selected solution concentration. Here, the tip slid on the calcite surface in a reciprocating motion, while the lateral deflection in the forward (trace) and backward (retrace) directions was acquired, with trace and retrace scans defining a single friction loop. Figure 5b–d shows representative friction loops, with a positive lateral force in trace and a negative lateral force in retrace over a selected displacement. The friction force was calculated by averaging over the half width of the trace and retrace scans and the standard deviation of the friction force was determined over ten loops. Replica experiments with different sliding lengths (100 nm, 200 nm and 1 μm) did not lead to any significant difference in the experimental results. The data shown in the main text correspond to a sliding length of 100 nm. The load-dependent friction-force measurements were performed at sliding velocities of 0.2 and 2 μm s$^{-1}$, which were maintained constant by a piezo motor. Here, the load was ramped up from 0.5 to 20 nN. In these experiments, the coefficient of friction ($\mu$) was obtained by calculating the slope of the friction vs. the load for each stress regime. The velocity-dependent friction-force measurements were conducted while the applied load was maintained constant; following loads were selected: 0.5, 1, 2 nN for LS regime, 5, 10, 15, 20 nN for IS regime, and 20, 30, 40 and 50 nN for HS regime. The sliding velocity was increased stepwise from 0.05 to 10 μm s$^{-1}$ at each selected load. The Hertzian average contact stress in each regime is 191 MPa (for 2 nN), 413 MPa (for 20 nN), and 733 MPa (for 50 nN), respectively (Supplementary Table 2). For both load- and velocity-dependent measurements, ten friction loops were measured at each velocity and each load. Replica experiments were performed at three different spots on each calcite sample to confirm reproducibility.

**Data availability**. All relevant data in this work are available from the authors per request.

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

## Acknowledgements

The authors would like to thank Prof. Dag Kristian Dysthe for his constructive suggestions and the fruitful discussion. The authors also thank Josue Lopez for providing the Matlab code to process the friction data. This work is supported by the National Science Foundation Grant CMMI-1435920.

## Author contributions

Y.D. performed experiments. R.M.E.-M. designed the research and supervised the graduate student. Y.D. and R.M.E.-M. analyzed the data and wrote the manuscript.

## Additional information

**Competing interests:** The authors declare no competing interests.

