## [Peer Review File · Nature Communications]

Reviewers' comments:

Reviewer #1 (Remarks to the Author):

Review of "About how water lubricates earthquakes" by Dia and Espinosa-Marzal.

In the paper the authors use lateral force microscopy to measure the friction as a silicon tip is moved across the surface of calcite in a fluid. The authors identify three distinct regions in their experiments and explain this behavior by invoking a series of different models. This paper is the first paper I am aware of that has used these methods to look at friction in geological materials and it has significant promise. But there are some problems, listed below, that need to be addressed before it can be published.

1. The paper is in need of significant editing. For a supposedly short paper, it is quite long (12 pages!). On the one hand the authors try to explain all their results (which makes the paper long); on the other hand the explanations are telegraphic (which makes the paper difficult to follow in parts). The authors do not reach the right balance.

2. There are a number of inaccuracies in the text, starting with the title: "About how water lubricates earthquakes". The paper does not deal with earthquakes; it deals with friction. Also water does not lubricate earthquakes; it potentially lubricates the faults that cause earthquakes. Also the paper doesn't explain the relationship between friction and lubrication and earthquakes. I think the title is an attempt to oversell the study.

3. Concerning the methods, the tip is made of silicon; the substrate is calcite. What interaction in nature is this suppose to simulate?

4. The tip radius is given as 100-150 nm but the manufactures give it as 8 nm. Why the difference?

5. The authors premise in the paper is that there is a direct relationship between nano-scale friction and the effective friction along faults during earthquakes. This relationship is definitely not obvious given the large spatial scales of faults and their complicated geometry.

6. Assuming that the three sliding regimes do occur in nature, what specific implications do they have for fault motion? What observations do they explain? This is completely unclear from the manuscript.

7. The explanation of the viscous shear model is unclear. It is peppered with jargon and phrases which the reader is assumed to be familiar with (e.g., shear-thinning). Why is the approximation in Equation 1 valid? I understand the argument but I think it's poorly written for a general scientific audience.

8. The authors explanation of the thermally activated mechanism is unclear. What are "thermally activated jumps"? What is the "humping velocity"? The authors need to explain this mechanism better.

9. The discussion of the pressure solution mechanism is another weak point. The authors state that their evidence is qualitative. Could they suggest how it might be tested quantitatively? I also found the argument on page 8 too convoluted to follow.

10. The overselling of the results is summarized by the final sentence: "The knowledge gained from this work is not only of importance in seismology, but also in other fields including carbonate rock weathering, mining, and enhanced oil recovery in petroleum industries." Perhaps. But how do the results actually affect these scenarios?

11. I would provide more specific comments on the text but the manuscript has no line numbers (or even page number for that matter). This is a shame.

Reviewer #2 (Remarks to the Author):

Dear Editor

The paper entitled "About how water lubricate earthquakes" by Y. Diao and M. Espinoza-Marzal, report on laboratory experiments that are designed to understand the deformation mechanism(s) of single crystal calcite. They explore the effect on applied stress, water solution and shear velocity on the frictional strength of single crystal calcite. They individuate three deformation regimes depending on the applied stress and highlight that at the highest stress regime and slowest velocity the strength of calcite is controlled by time-dependent pressure solution creep that induce a weakening effect on the frictional strength. The experiments are somewhat interesting, even if this effect was previously documented in rock deformation experiments (i.e. Carpenter et al., 2016 GJI), and the authors do not consider this work. Overall the paper is not well organized. The introduction is vague and the connection with earthquakes is not clear and somewhat incorrect. Starting from the title, it is possible to understand that the authors do not have a background in Geology, since water lubricates fault and not earthquakes, that indeed are the result of sliding on faults. Throughout the text there are many big conceptual errors when referring to geological concepts, with specific geophysical terminology used improperly most of the time (see attached file with comments). The authors try to make a connection with slow earthquakes that seems to drop from the sky, without explaining what is the physical mechanism that they are trying to understand. The methodology is also somewhat vague. They describe properly the boundary conditions of the experiments (only regarding the water solution and the applied load) but there is a total lack on experimental techniques. I would have expected, at least in the supplementary material, a schematic of the apparatus and some typical experimental curves. Without showing the apparatus they are using, who, like me, is not familiar with it cannot understand how the experiments were performed making the interpretation of the results quite difficult. The results are the only part that is well organized and described, even if the figures are not of a very high quality and the data are difficult to visualize due to the vary small panels (e.g. Fig 1 and 4). However, the biggest problem of this manuscript, in my opinion, is the total lack of discussions. The authors dedicate as small paragraph to the discussion where they make some statement without any connection with the data they present. Overall the paper is a fine data report but nothing more. To conclude, I feel like the paper needs a complete rewriting of the introduction and since there are not discussion and conclusion relevant to the community, they should write them from scratch. For these reasons, I suggest rejecting the paper. For further details please see the file attached with the comments (I would have written a line by line comment letter but there is no line numbering on the manuscript).

Sincerely

Editorial Note: In their review of the first version of this manuscript, reviewer 2 added their comments to the manuscript file. These comments, excluding minor textual revisions, have been copied into this Peer Review File.

Abstract, 1st line: This sentence is confusing since two adjacent tectonic plates undergoing shear already represent a fault.

Abstract, 2nd line, 'this knowledge': What knowledge?

Abstract, 5th line, 'slow earthquakes': I don't understand the connection with slow earthquakes here. The authors should write something since slow earthquakes does not have only one velocity and it is not contextualized what is a slow earthquake and seismic activity.

First paragraph, 6th line, 'AFM': No defined. what does it mean?

First paragraph, 12th line, 'mantle': Why mantle? In the mantle, rocks are already in the ductile deformation regime, while earthquakes take place in the brittle crust at depth usually less than 15km. Only in subduction zones they can arrive at greater depth.

First paragraph, 13th line, 'process': It is conventionally called seismic cycle

Second paragraph, 3rd line, Reference 10: There are many references here from the rock mechanics literature that have demonstrated this. Please add some meaningful references here.

Third paragraph, 2nd line, 'subduction fault': this term does not mean anything or at least should be more specific. One possible change would be "subduction faulting".

Third paragraph, 5th line, Reference 19: "to alter the magnitude and frequency of earthquakes" I don't think that there are any studies that can prove what is said in this sentence since it is still a timely problem. Particularly, the control of water on magnitude and frequency of earthquakes is quite far to be understood (for example McGarr, 2014). The reference cited here only refers to dynamic weakening processes that take place during rupture propagation and lubricate the fault. So this statement is rather incorrect.

Third paragraph, 6th line, References 20, 21: In these references you have neglected: Carpenter, B.M., Collettini, C., Viti, C., Cavallo, A., 2016. The influence of normal stress and sliding velocity on the frictional behaviour of calcite at room temperature: Insights from laboratory experiments and microstructural observations. *Geophys. J. Int.* 205, 548–561. doi:10.1093/gji/ggw038, where the author clearly show very similar results as in the present manuscript.

Third paragraph, 9th line, 'Inspiring as it is': Don't really understand what does it mean

Fifth paragraph, 1st line: It would be nice to see a schematic of the apparatus in the supplementary material.

Sixth paragraph, 2nd line: How can these values be compared with existing rock physics experiments in order to better understand these processes at a larger scale? In particular because the shear

velocity explored in this study are very similar to the common velocities used in rock deformation experiments.

Seventh paragraph, 1st line, 'friction force as a function of velocity': There is no description either here or in the supplementary material on how these tests were performed, making difficult to interpret the data. For example how velocity was controlled and how was increased, stepwise?

Figure 1: What is the orange and red curve in these panels? Those colors are not indicated in the legend. I guess they represent the data showed on the top panel but it should be mentioned somewhere. Compare fig F and J and you the same friction trends??

Eighth paragraph, 3rd line, 'FL': FL is introduced here but never defined. From Gnecco et al., 2000 I can see that it is the Lateral Force. Is this correct?

Figure 2: I would expect some more systematic changes of lambda as a function of concentrations

Ninth paragraph, 10th-11th lines: This sentence is not clear to me. It should explain the relation between F^* and E_0 ?

Twelfth paragraph, 6th line, 'slip is facilitated at the highest applied loads': This is not necessarily true since only for the case of $v = 0.2\mu\text{m/s}$ there is a weakening in friction, otherwise friction is equal or higher. So I think it is only the combination of high-load and low-velocity to induce weakening. This was also documented in Carpenter et al., 2016.

Figure 4: Most of the curves looks to be S-shaped even in a log-log space. Why don't you show them in a linear-log space as in the inset?

Thirteenth paragraph, 9th line: cannot it be simply time-dependent contact growth induced by fluids and higher load? Or some plastic/dislocation processes? If this is the only evidence for pressure solution it is rather weak. How long were the experiments? Since it is not specified, but I can infer you slid for 100nm at $0.2\mu\text{m/s}$..

Thirteenth paragraph, 14th line: These evidences are indicative but not directly related to pressure solution. I would like to see some images of the grain at the end of experiment to be fully certain.

Nineteenth paragraph, 2nd-3rd line: However, the rocks that characterize these plate boundary faults are totally different from pure calcite. In fact in both the mentioned plate boundary fault phyllosilicates are the most abundant mineralogical phase present.

Nineteenth paragraph, 6th line, 'two fault planes': Please see the definition of fault plane, if you say two fault planes it means that you are considering two different faults. A fault plane is characterized by a discontinuity in the crust along which shear is localized. There are two fault surfaces and one plane.

Nineteenth paragraph, 7th line: I don't think that it is so simple since seismogenic faults are greatly influenced by Temperature and many other factors.

Reviewer #3 (Remarks to the Author):

This is an extremely thorough and thoughtful paper and provides nanoscale insight into geological processes. I recommend publication. The observations are novel and the conclusions appear justified. While it may appear at first sight that nanoscale measurements with AFM are inappropriate to describe geological events, the local pressures are in fact relevant and comparable. The data is of extremely high quality and has been analysed with care. In terms of relevance, it is a joy to see fundamental physical chemistry being rigorously applied to explain geological phenomena.

The paper is clear and well written.

While the pressures and compositions are relevant, temperature is not mentioned. It is not necessary to show temperature dependent data, it behooves the authors to address the relevance of the temperatures of their experiments in the context of the seismic processes they purport to model.

Minor comments:

The statement that calcium does not populate mica needs perhaps to be justified: there are experimental studies that strongly imply population. (this is not central to the paper but is relevant to the general credibility)

The use of the word "atop" on page 8 seems to be a typo.

It might be politic to find an alternative expression to "humping velocity" on page 4.

Mark Rutland

Dear editor and reviewers,

Thank you very much for all your comments. We have addressed each comment point by point and have added our responses to this document and highlighted the changes in the manuscript. To answer some of the comments we have conducted additional reference measurements that have been added to the SI (FIGs 6, 7, 9, 11 and 12). We hope that you will find the modified versions of manuscript and SI to be appropriate for publication. Thanks again for this opportunity!

Reviewers' Comment Reviewer

Reviewer #1 (Remarks to the Author):

Thank you very much for your critical comments. We have reviewed the manuscript according to your comments and corrections. We believe that your revision has improved remarkably our manuscript and are very thankful.

Review of “About how water lubricates earthquakes” by Diao and Espinosa-Marzal.

In the paper the authors use lateral force microscopy to measure the friction as a silicon tip is moved across the surface of calcite in a fluid. The authors identify three distinct regions in their experiments and explain this behavior by invoking a series of different models. This paper is the first paper I am aware of that has used these methods to look at friction in geological materials and it has significant promise. But there are some problems, listed below, that need to be addressed before it can be published.

1. The paper is in need of significant editing. For a supposedly short paper, it is quite long (12 pages!). On the one hand the authors try to explain all their results (which makes the paper long); on the other hand the explanations are telegraphic (which makes the paper difficult to follow in parts). The authors do not reach the right balance.

We have tried to explain better our results and have moved some results to the supplementary information. All the additions still keep the number of words below the given limit (<5000 words). We hope that we have reached the right balance yet.

2. There are a number of inaccuracies in the text, starting with the title: “About how water lubricates earthquakes”. The paper does not deal with earthquakes; it deals with friction. Also water does not lubricate earthquakes; it potentially lubricates the faults that cause earthquakes. Also the paper doesn't explain the relationship between friction and lubrication and earthquakes. I think the title is an attempt to oversell the study.

Thank you very much for this criticism. We think that a more appropriate title is “About how water lubricates faults”. The relation between friction and lubrication (called hydration lubrication) is very important and we have added an explanation:

Line 101: “Our previous studies by AFM¹ allowed us to gain molecular insight into the aqueous solution confined between calcite and a (micron-sized) silica tip by scrutinizing the normal surface forces down to subnanometer separations. A strong repulsive force –termed hydration repulsion²⁻⁴ was measured between calcite and silica, and it prevented the solution to be squeezed-out under a normal stress (here, smaller than 200 MPa). The nanoconfined solution is composed of water layers that are strongly adsorbed to the calcite surface and located underneath a layer of partially dehydrated calcium ions, and fully hydrated calcium ions further away from the surface.”

Line 119: “This suggests that the friction force in the LL-regime is mainly due to a viscous force that results from shearing the confined aqueous brine, with no significant influence of the concentration. The lubrication mechanism provided by the confined solution has been previously called “hydration lubrication”⁵.

In this version of the manuscript we emphasize less the relation to earthquake ruptures. Nevertheless, there is certain connection, which is described as follows:

Line 22: “When the static friction between two adjacent tectonic plates undergoing shear is overcome, fault deformation occurs, and, under conditions of unstable slip, an earthquake rupture may take place⁶”.

Line 353: “Calcite-rich rocks have also shown unstable frictional behavior in laboratory experiments under conditions pertinent to earthquake nucleation in nature⁷⁻¹⁰. Aging effects, as considered in RSF laws, can result from slow creep of asperity microcontacts under compression¹¹ and from chemical bonds forming across the interface¹²⁻¹⁶. Such aging effects can cause a velocity-weakening frictional strength, and under, specific conditions, unstable sliding according to RSF laws; while velocity-strengthening friction, like the one observed here, is characteristic of stable fault dynamics and it should favor aseismic creep¹⁷. According to our work, the effect of aging on the frictional strength of the calcite-silica single-asperity contact is negligible, and hence, stable slip should be expected. However, it has been shown that fluid pressurization in carbonate-bearing fault gouge may trigger a dynamic instability even in case of velocity-strengthening friction¹⁸. Despite gouge compaction caused by pressure solution was observed at low fluid pressures, high fluid pressures led to significant dilation, which resulted in fast acceleration and earthquake slip. The authors argued that RSF laws cannot describe the effect of the fluid overpressure, and proposed a change of the model to account for it. While applying this model to the single-asperity contact is out of the scope of this work, our proposed mechanism for the remarkable weakening at the nanoscale could explain Scuderi et al.’s observations¹⁸. This let us hypothesize that earthquake ruptures might be triggered or propagated by fault weakening induced by pressure solution of calcite asperity contacts.”

3. Concerning the methods, the tip is made of silicon; the substrate is calcite. What interaction in nature is this supposed to simulate?

This is an important point that we address in detail now in the new manuscript:

1) Although the tip bulk material is silicon, silicon is naturally oxidized in contact with ambient air and in water. It has been shown that the thickness of the silica layer is ~ 3 nm.¹⁹ As a result, the tip surface and the thin region beneath the surface is amorphous silica, likely hydroxylated upon immersion in water.

2) Amorphous silica can be found in nature (e.g. opal). Nevertheless, it has been demonstrated that the frictional characteristics of silica glasses and crystalline (quartz) and other silicates is similar. Hence, the system represents a calcite-silicate contact in nature. Note that this analogy has been performed before for silica-silica contacts^{12,13}.

3) We have conducted additional measurements on reference systems to evaluate the influence of the tip on the frictional response. We have investigated friction as a function of speed for a) a naturally oxidized silicon tip (i.e. the silicon tip has a native oxide layer, and it is abbreviated as silica tip in the following) sliding along a naturally oxidized silicon (i.e. silica) substrate and b) a silicon nitride tip sliding on a calcite surface. We show (a) that a silica-silica surface has a completely different behavior and (b) that the use of a silicon nitride on calcite yields similar results to those described in the manuscript. We show the results of these reference experiments in the SI and have added following explanation to the manuscript:

Line 190: “It is important to briefly discuss the influence of the AFM tip on these results. First, we note that analogous results are obtained with a silicon nitride AFM tip (FIG. S6), and hence, the bulk material of the tip do not play any role on the observed mechanisms of energy dissipation. Second, the friction force measured with a naturally oxidized silicon tip sliding on a silica surface decreases with increase in velocity (FIG. S7), in contrast to the results shown in FIG. 1. This velocity-weakening friction force of silica-silica interfaces has been attributed to the formation of hydrogen and/or siloxane bonds across the interface, which causes contact aging^{13,20}. Therefore, the thermally activated slip underlying the frictional strength of the single-asperity contact of this work mainly reflects the frictional characteristics of the calcite surface.”

4. The tip radius is given as 100-150 nm but the manufactures give it as 8 nm. Why the difference?

Thanks for emphasizing the difference since it is also key for this study. Note that this high tip radius is not unique to our work; e.g. see recent paper by Carpick¹³ and Salmeron's work²⁰. It has been shown that the tip can break when it hits the surface due to its small radii and the large pressure²¹. Hence, the tip radius that we report here is that after some pre-sliding tests to avoid that the tip breaks during our experiments and to ensure that it remains stable. As a side note, we have observed a variation of radii that are larger than given by the manufacturer's specifications, and therefore, we routinely image the tips before and after the experiments.

Second, we have confirmed that the results are similar when measured with different tip radii, as also other studies have shown earlier²². However, the applied pressures at the same load are higher with sharper tips and that is a key point for our work since the pressure determines the transition between the different regimes, since it leads to a change of the behavior of the confined solution. To be able to probe hydration lubrication (at small pressures), we need to apply very small pressures (and thus, loads), which is facilitated if we use larger tip radii due to the instrumental limitation to apply loads smaller than 0.2 nN with high precision. These are the reasons for the selection of this range of radii from 100 to 150 nm.

We have added following clarification to the paper:

Line 66: "Although it is known that the mechanisms of energy dissipation do not depend on the size of the non-adhesive contact²², several replica experiments were conducted with different tip sizes to confirm the reproducibility of results. Since fluid overpressures cannot be applied in our setup, and considering that the smallest reliable force that the instrument can apply is ~0.2 nN, large tip radii were selected to allow applying pressures below the osmotic pressure of the confined fluids²³. The tip radius of the reported experiments thus ranges from 100 nm to 150 nm, as determined by scanning electron microscopy (SEM) imaging (FIG. S2). SEM imaging of the AFM tips before and after friction force measurements showed no significant changes in tip size, thereby excluding any irreversible change of the AFM tip, e.g. by wear or plastic deformation, during the duration of the experiment."

5. The authors premise in the paper is that there is a direct relationship between nano-scale friction and the effective friction along faults during earthquakes. This relationship is definitely not obvious given the large spatial scales of faults and their complicated geometry.

The relation across scales was not clear in the manuscript. The nanoscale friction force measurements represent the frictional response of a single asperity, which has a radius of 100-150 nm in the reported results. Nevertheless, it has been demonstrated²² that the mechanisms of energy dissipation underlying the friction force (i.e. the origin of friction) do not depend on the size of the asperity for a non-adhesive single-asperity contact, like the one reported here. In those reported works, friction force measurements were conducted on various substrates (silica and mica), bare and coated with monolayers, smooth and rough, carried out by SFA (contact area with diameters of μm -size) and by AFM (nm-sized contacts). They confirmed the correspondence of the coefficient of friction by comparing results obtained on non-adhesive single-asperity contacts with diameters that differed over more than 3 orders of magnitude.

In nature, microscopic and submicroscopic asperities are present on rock surfaces due to surface roughness (over several length scales) and in fault gouges, due to the grain size distribution. For instance, particle size distribution (PSD) measurements revealed the grain size of fault gouge in earthquake rupture zones to be 0.04-200 μm , whereas they can become smaller than 1 μm after long-run disaggregation²⁴. The frictional strength of faults is determined by multi-asperity contacts.

To enable a comparison across different scales, we have included now the range of normal stresses for each regime to the manuscript, since it is the normal stress, which determines the mechanism underlying friction.

We have added a short paragraph to the manuscript to clarify this comparison:

Line 91: "Importantly, this range of normal stresses is relevant to that along faults in the upper Earth crust, which is in the order of 100 MPa²⁵, but higher at single-asperity contacts, which determine the frictional

strength of faults. The confinement geometry provided by the AFM allows examining the slip phenomena at the scale of a single asperity.”

6. Assuming that the three sliding regimes do occur in nature, what specific implications do they have for fault motion? What observations do they explain? This is completely unclear from the manuscript. Thanks so much for this comment. We recognize that we had not explained this in depth and have added a discussion with a series of implications at the end of the manuscript. Figure 5 has been also modified so that the connection to fault dynamics is easier (we use the friction force normalized by the applied normal load) to illustrate the three regimes, and write:

Line 294: “In summary, by measuring the friction as a function of load and sliding velocity with an AFM tip sliding along a calcite surface in calcium chloride brines of different compositions, we resolved three different mechanisms responsible for the frictional strength of this nanoscale single asperity (FIG. 5a): hydration lubrication at normal stresses (σ_n) <200 MPa, in which the confined brine lubricates the nanoscale contact and the frictional strength is dictated by the viscous shear force of the fluid film; velocity-strengthening friction at higher normal stresses (σ_n <400 MPa), as a result of a thermally activated slip of the surface-localized hydrated ions upon shear, strongly dependent on the solution composition; and pressure-solution facilitated slip which re-activates hydration lubrication at higher stresses (σ_n >400 MPa) and sufficiently small sliding velocities and causes a significant weakening of the frictional strength. The latter was more pronounced with increasing calcite surface reactivity, which was achieved by increasing the calcium concentration in this work; in nature, it will depend on the specific composition of the brine. “

Please, see more details in the caption of FIG. 5 and in the discussion.

7. The explanation of the viscous shear model is unclear. It is peppered with jargon and phrases which the reader is assumed to be familiar with (e.g., shear-thinning). Why is the approximation in Equation 1 valid? I understand the argument but I think it’s poorly written for a general scientific audience.

Thank you, this has been modified. See new explanation:

Line 97: “The linear relationship in the log-log scale indicates that the friction force follows a power law. This is reminiscent of the shear response of fluids as described by the Ostwald-de Waele equation $\sigma \sim (\dot{\gamma})^n$, being σ the shear stress, $\dot{\gamma} = v/D$ the shear rate, D the fluid film thickness, and n an exponent²⁶. Our previous studies by AFM¹ allowed us to gain molecular insight into the aqueous solution confined between calcite and a (micron-sized) silica tip by scrutinizing the normal surface forces down to subnanometer separations. A strong repulsive force –termed hydration repulsion²⁻⁴ – was measured between calcite and silica, and it prevented the solution to be squeezed-out under normal stress (here, smaller than 200 MPa). The nanoconfined solution is composed of water layers that are strongly adsorbed to the calcite surface and located underneath a layer of partially dehydrated calcium ions, and fully hydrated calcium ions further away from the surface.”

Line 115: “The fitting parameters are quite similar under all loading conditions and concentrations, averaging as $F_L = (0.0167 \pm 0.007)v^{(0.38 \pm 0.06)}$. This suggests that the friction force in the LL-regime is mainly due to a viscous force that results from shearing the confined aqueous brine, with no significant influence of the concentration. The lubrication mechanism provided by the confined solution has been previously called “hydration lubrication”⁵.”

8. The authors explanation of the thermally activated mechanism is unclear. What are “thermally activated jumps”? What is the “humping velocity”? The authors need to explain this mechanism better.

Thank you. This has been revised and now it reads:

Line 130: FIGs. 1. f-j show the friction force as a function of velocity in the selected CaCl₂/CaCO₃ solutions in the IL-regime ($\sigma_n \sim 200$ -400 MPa). The friction force follows an approximately linear relationship with the logarithm of velocity, i.e. $F_L \sim \log(v)$, thereby indicating a pathway of energy dissipation (friction) that

differs from the viscous dissipation at smaller normal stresses. The logarithmic dependence of the friction force is attributed to a shear-promoted thermally activated slip in the context of transition state theory²⁷. Based on our previous work¹, we assume that (hydrated) calcium ions remain localized on oppositely charged surface sites under high applied pressures. Upon shear, the net frequency at which the hydrated ions can overcome the activation energy (ΔE) to crossing to an adjacent energy minimum is increased by the applied lateral force⁵. This yields the following expression for the friction force F_L :

$$F_L = \frac{\Delta E}{\lambda} + \frac{k_B T}{\lambda} \log(v/v_0) \quad \text{Eq. 1}$$

k_B being the Boltzmann constant, T the absolute temperature, v_0 a reference attempt velocity, and λ is the coherence length, also called the stress-activation length. The coherence length gives the length through which the applied shear stress reduces the energy barrier ΔE , i.e. the distance from the ion's initial stable position to the top of the activation barrier, the transition state, before it falls to the next energy minimum. The solid lines in FIGs. 1. f-j show the fits to Eq. 1; the fitted coherence length λ for all selected concentrations is plotted in FIG. 2a. The coherence length, λ , gives thus the length over which the confined ions and water are capable of building up strain in response to shear. As shown in FIG. 2a, λ decreases significantly with increase in normal load at all investigated concentrations, thereby reflecting the decrease in mobility of the confined solution with applied load.”

9. The discussion of the pressure solution mechanism is another weak point. The authors state that their evidence is qualitative. Could they suggest how it might be tested quantitatively? I also found the argument on page 8 too convoluted to follow.

Thank you very much for motivating us to explore more this phenomenon. We have improved this part in a number of ways. The best method to evaluate pressure solution is to image the track of the tip after the friction force measurements. We have completed additional friction measurements in the brine to prove that pressure solution happens by imaging the track of the tip. We calculated the volume of the track and were able to roughly estimate a dissolution rate of $1.5 \cdot 10^{-10}$ mol/cm²/s, which is in the same order of magnitude as the dissolution rate of stress-free calcite surface. It is reasonable to expect that dissolved ions will re-precipitate back into the track concurrently with the sliding of the tip moving away, which would justify an underestimation of the dissolution rate. Therefore, we have concluded that the imaging of the track of the tip does provide convincing evidence that pressure solution takes place during the friction force measurements. See the additional measurements in the SI (FIG S11 and S12).

The main text now says:

Line 223: “At the sliding conditions at which the F_L vs. L -slope was observed to decrease, imaging of the calcite surface showed the track of the tip (FIG. S11), while re-precipitation sometimes occurred close to the track (see the bright spot in FIG. S11c). Images taken after similar friction measurements in ethanol, in which calcite is insoluble, do not show any change of the surface, which excludes that brittle fracture and plastic deformation of calcite could be responsible for the observed track in the presence of the brine (FIG. S12). Further, we note that a similar decrease of friction with load was not detected for silica-silica contacts in aqueous solution (FIG. S7) and AFM imaging did not show any change of the silica surface and of the tip. Hence, the remarkable drop in friction with increase in load (FIG. 3) is attributed to the activation of the pressure solution of calcite at sufficiently high normal stresses and small sliding velocities.”

Further the discussion about the dissolution kinetics in the main text has been simplified:

Line 269: “The scrutiny of friction as a function of velocity (FIG. 4) is insightful, since it can be related to the kinetics of pressure solution of calcite. Taking a normal stress $\sigma_n \sim 540$ MPa (20 nN), the thermodynamic potential $\Delta\mu$ that drives the dissolution of the stressed calcite is given by

$\Delta\mu/kT = V_m\sigma_n/kT = \log(a^p/a^0) \sim 7.87$, V_m being the molecular volume of calcite, and a^0 and a^p the activity of the solution in equilibrium with stress-free and stressed calcite, respectively.²⁸ The chemical potential difference induced by σ_n leads to a sudden increase of the local solubility as well as a dramatic undersaturation of the interfacial solution. The reaction-limited dissolution rate under such conditions is proportional to the solution activity.²⁹ Thus, the dissolution rate under normal stress is estimated as $r^p \sim r^0 a^p/a^0$, being r^0 the dissolution rate of stress-free calcite. Pressure solution might be also electrochemically enhanced due to a difference in surface potential between the confining surfaces³⁰, but this effect was neglected as first approximation.

Let us now take $r^0 \sim 1.8 \cdot 10^{-10}$ mol/cm²/s as the reaction-limited dissolution rate of calcite in water³¹, which is equivalent to ~ 1 ion pair/nm²/s. Upon the assumed normal stress, the rate increases to $r^p \sim e^{7.87}$ ion pair/nm²/s. Given the unit cell for calcite to be $c \sim 0.5$ nm, the time required for an ion pair to be dissolved under the normal stress is $\Delta t = 1/c^2 r^p \sim 1.5 \times 10^{-3}$ s. Taking $a = 6.8$ nm as the estimated diameter of the Hertzian contact area, it requires a velocity smaller than $v = \frac{a}{\Delta t} = 4.43$ $\mu\text{m/s}$ for an ion pair to be pressed by the AFM tip longer than Δt . Theoretically, a sliding event with either a smaller velocity (to allow a longer tip-substrate contact time), or a higher stress (to enhance the dissolution rate further) should let pressure solution happen while the tip slides. The same estimation practice at the loads of 10 nN and 50 nN results in 0.7 $\mu\text{m/s}$ and 94 $\mu\text{m/s}$, which agrees very well with the range of applied velocities in our study. We thus conclude that pressure solution of calcite decreases friction by promoting hydration lubrication under high normal stress, thereby facilitating slip at the single-asperity contact. While ion diffusion does not play a relevant role at the level of a nanoscale asperity, pressure solution creep of rocks in nature and fault gouges might require smaller sliding velocities due to the additional retarding effect of ion diffusion.”

10. The overselling of the results is summarized by the final sentence: “The knowledge gained from this work is not only of importance in seismology, but also in other fields including carbonate rock weathering, mining, and enhanced oil recovery in petroleum industries.” Perhaps. But how do the results actually affect these scenarios?

You are right. We have removed this sentence and focus only on the implications for seismology. Please, see the corrected version of the discussion.

11. I would provide more specific comments on the text but the manuscript has no line numbers (or even page number for that matter). This is a shame.

We sincerely apologize for the missing page and line numbers. They have been added now.

Reviewer #2 (Remarks to the Author):

Dear Editor,

The paper entitled “About how water lubricate earthquakes” by Y. Diao and M. Espinosa-Marzal, report on laboratory experiments that are designed to understand the deformation mechanism(s) of single crystal calcite. They explore the effect on applied stress, water solution and shear velocity on the frictional strength of single crystal calcite. They individuate three deformation regimes depending on the applied stress and highlight that at the highest stress regime and slowest velocity the strength of calcite is controlled by time-dependent pressure solution creep that induce a weakening effect on the frictional strength. The experiments are somewhat interesting, even if this effect was previously documented in rock deformation experiments (i.e. Carpenter et al., 2016 GJI), and the authors do not consider this work.

Overall the paper is not well organized. The introduction is vague and the connection with earthquakes is not clear and somewhat incorrect. Starting from the title, it is possible to understand that the authors do not have a background in Geology, since water lubricates fault and not earthquakes, that indeed are the result of sliding on faults.

Throughout the text there are many big conceptual errors when referring to geological concepts, with specific geophysical terminology used improperly most of the time (see attached file with comments).

The authors try to make a connection with slow earthquakes that seems to drop from the sky, without explaining what is the physical mechanism that they are trying to understand.

The methodology is also somewhat vague. They describe properly the boundary conditions of the experiments (only regarding the water solution and the applied load) but there is a total lack on experimental techniques. I would have expected, at least in the supplementary material, a schematic of the apparatus and some typical experimental curves. Without showing the apparatus they are using, who, like me, is not familiar with it cannot understand how the experiments were performed making the interpretation of the results quite difficult.

The results are the only part that is well organized and described, even if the figures are not of a very high quality and the data are difficult to visualize due to the vary small panels (e.g. Fig 1 and 4).

However, the biggest problem of this manuscript, in my opinion, is the total lack of discussion. The authors dedicate as small paragraph to the discussion where they make some statement without any connection with the data they present. Overall the paper is a fine data report but nothing more.

To conclude, I feel like the paper needs a complete rewriting of the introduction and since there are not discussion and conclusion relevant to the community, they should write them from scratch. For these reasons, I suggest rejecting the paper. For further details please see the file attached with the comments (I would have written a line by line comment letter but there is no line numbering on the manuscript).

Sincerely

Dear reviewer,

Thank you very much for your revision and for the immense input. We agree with you that the manuscript required significant re-writing concerning the introduction and the discussion. We have extended the literature revision, which has also enormously helped to improve the manuscript. We have added a description of the method to the SI, as well as additional reference experiments to support our findings. We hope that you find our manuscript now suitable for publication. Also, we apologize because we forgot to add the line numbers (they have been added now). We thank you very much for your insightful review. Please, find below the answers to each of your comments and the corresponding revision.

Page 1

“About how water lubricates earthquakes”.

Comment Reviewer: Water can lubricate faults that is where earthquakes nucleate and not the opposite.

Thank you very much. We have modified the title, also following the editor’s wish, which agrees with you. It now reads:

“About how water lubricates faults”.

Abstract

1. “The friction between two adjacent tectonic plates undergoing shear in a fault may dictate seismic activities”.

Comment Reviewer: This sentence is confusing since two adjacent tectonic plates undergoing shear already represent a fault.

This has been corrected to:

Line 9: “The friction between two adjacent tectonic plates undergoing shear may dictate seismic activities, and hence, understanding mechanisms underlying the frictional strength of fault rocks is of paramount relevance.”

In this study, we aim to advance this knowledge by investigating the frictional characteristics of calcite in aqueous environment, representing a common fault type in nature, i.e. carbonate rock in presence of a brine.

Comment Reviewer: what knowledge

This has been corrected to:

Line 11: “In this study, we investigate the frictional characteristics of calcite in aqueous environment”.

By conducting nanoscale lateral force microscopy at sliding velocities similar to the slip rates in slow earthquakes, ...

Comment Reviewer: I don't understand the connection with slow earthquakes here. The authors should write something since slow earthquakes does not have only one velocity and it is not contextualized what is a slow earthquake and seismic activity.

We agree and have removed the comments about slow earthquakes from the manuscript.

Main text

When the static friction between two adjacent tectonic plates undergoing shear in a fault is overcome, an earthquake rupture takes place, thereby releasing an immense amount of energy.

Comment Reviewer: ~~in a fault~~

This has been corrected to:

Line 22: “When the static friction between two adjacent tectonic plates undergoing shear is overcome, fault deformation occurs, and, under conditions of unstable slip, an earthquake rupture may take place”.

For instance, the static friction force between a sharp AFM tip and silica was evaluated in light of rate and state friction (RSF) laws, as typically used to describe the frictional behavior of fault gouges.

Comment Reviewer: No defined. what does it mean?

It has been added that AFM is the abbreviation for Atomic Force Microscopy in line 32.

Further, AFM studies on anisotropic mineral surfaces, such as antigorite, dolomite and calcite, showed that both stick-slip behavior and kinetic friction force depend on the sliding direction, which suggests that, at the geological scale, mantle rocks may exhibit opposite frictional behaviors during a seismic process.

Comment Reviewer: Why mantle? In the mantle, rocks are already in the ductile deformation regime, while earthquakes take place in the most particular crust at depth usually less than 15km. Only in subduction zones they can arrive at greater depth.

Comment Reviewer: It is conventionally called seismic cycle

Thank you for the correction. The authors in that paper discussed subduction, and mentioned that the factors controlling the frictional strength are related to the effect of water. They mentioned the forearc serpentinized mantle and the high temperature. Nevertheless, we have removed this description since it is not relevant for our work.

Thanks, we have modified the term to seismic cycle.

Carbonate rocks, like calcite, abound in lithosphere and are comparatively reactive in contact with aqueous brines [Ref 10].

Comment Reviewer: There are many references here from the rock mechanics literature that have demonstrated this. Please add some meaningful references here.

Thank you. We have added several references about carbonate rocks and calcitic fault gouges :

Verberne, B. A., He, C. R. & Spiers, C. J. Frictional Properties of Sedimentary Rocks and Natural Fault Gouge from the Longmen Shan Fault Zone, Sichuan, China. *B Seismol Soc Am* **100**, 2767-2790, doi:10.1785/0120090287 (2010).

Carpenter, B. M., Collettini, C., Viti, C. & Cavallo, A. The influence of normal stress and sliding velocity on the frictional behaviour of calcite at room temperature: insights from laboratory experiments and microstructural observations. *Geophysical Journal International* **205**, 548-561, doi:10.1093/gji/ggw038 (2016).

Royne, A., Bisschop, J. & Dysthe, D. K. Experimental investigation of surface energy and subcritical crack growth in calcite. *J Geophys Res-Sol Ea* **116**, doi:Artn B04204 10.1029/2010jb008033 (2011).

Violay, M. *et al.* Pore fluid in experimental calcite-bearing faults: Abrupt weakening and geochemical signature of co-seismic processes. *Earth and Planetary Science Letters* **361**, 74-84, doi:<https://doi.org/10.1016/j.epsl.2012.11.021> (2013).

Carpenter, B. M., Scuderi, M. M., Collettini, C. & Marone, C. Frictional heterogeneities on carbonate-bearing normal faults: Insights from the Monte Maggio Fault, Italy. *Journal of Geophysical Research: Solid Earth* **119**, 9062-9076, doi:10.1002/2014JB011337 (2014).

Tesei, T. *et al.* Friction and scale-dependent deformation processes of large experimental carbonate faults. *Journal of Structural Geology* **100**, 12-23, doi:<https://doi.org/10.1016/j.jsg.2017.05.008> (2017).

Scuderi, M. M., Collettini, C. & Marone, C. Frictional stability and earthquake triggering during fluid pressure stimulation of an experimental fault. *Earth and Planetary Science Letters* **477**, 84-96, doi:<https://doi.org/10.1016/j.epsl.2017.08.009> (2017).

Page 2

During subduction fault, that is, when one tectonic plate moves beneath another, water can remain trapped at the plate boundaries, which may lower the mechanical coupling between plates, thereby reducing the frictional resistance.

Comment Reviewer: this term does not mean anything or at least should be more specific. One possible change would be "subduction faulting".

Thanks. We have corrected the description to:

Line 42: “While RSF models predict that fluid overpressure should stabilize fault slip in subduction zones, recent seismic studies suggest that the pore fluid can allow slip to occur along an otherwise frictionally unfavorable fault³², which is also supported by laboratory experiments¹⁸..”

Water has thus been proposed to serve as a “lubricant” between plates and to alter the magnitude and frequency of earthquakes [Ref 19].

Comment Reviewer: "to alter the magnitude and frequency of earthquakes" I don't think that there are any studies that can prove what is said in this sentence since it is still a timely problem. Particularly, the control of water on magnitude and frequency of earthquakes is quite far to be understood (for example McGarr, 2014). The reference cited here only refers to dynamic weakening processes that take place during rupture propagation and lubricate the fault. So this statement is rather incorrect.

Thank you. We have corrected this statement and added new references:

Line 41: “Water may play a significant role on earthquake ruptures and several mechanisms have been proposed to explain this phenomenon³²⁻³⁴. While RSF models predict that fluid overpressure should stabilize fault slip in subduction zones, recent seismic studies suggest that the pore fluid can allow slip to occur along an otherwise frictionally unfavorable fault³², which is also supported by laboratory experiments¹⁸. The latter show that the presence of water may lead to a long-term weakening of fault strength caused by microfracturing of asperities as a result of subcritical crack growth induced by chemical reaction of water with the strained apexes of cracks³⁵⁻³⁷. For instance, the fracture surface energy of calcite becomes 2–3 times lower compared to that of dry calcite³⁸. Further, velocity-weakening friction and unstable stick-slip was observed for limestone gouge saturated with water at high temperature (150°C), consistent with crystal plasticity at asperity contacts³⁹. An additional weakening mechanism of wet fault gouges is proposed to rely on intergranular lubrication facilitated by water adsorption⁴⁰. Finally, the frictional strength of calcitic gouge has been shown to transition from brittle behavior and unstable sliding to semi-brittle behavior – characterized by a velocity-strengthening frictional strength – when decreasing the sliding velocity. The authors attributed the semi-brittle behavior to fluid-assisted pressure solution of calcite at sufficiently high applied stresses and low speeds accompanied by precipitation of stress-free nanogranular calcite, thereby causing an overall compaction of the fault gouge⁸.”

Previous microscopic studies [Ref. 20, 21] on calcite-rich and quartz fault gouges saturated with water proposed the frictional behavior to be governed by multiple mechanisms, including pressure solution accommodated sliding, cracking, compaction, crystal plasticity, and intergranular lubrication due to water adsorption.

Comment Reviewer: In these references, you have neglected:

Carpenter, B.M., Collettini, C., Viti, C., Cavallo, A., 2016. The influence of normal stress and sliding velocity on the frictional behaviour of calcite at room temperature: Insights from laboratory experiments and microstructural observations. *Geophys. J. Int.* 205, 548–561. doi:10.1093/gji/ggw038

Where the author clearly show very similar results as in the present manuscript.

Thank you. We have greatly enjoyed reading this paper, whose reference has been added (see introduction and discussion) due to its relevance to our work.

Inspiring as it is, the investigated system was too complex for each mechanism to be elucidated and the micrometer length scale did not allow one to probe the molecular details.

Comment Reviewer: Don't really understand what does it mean.

This sentence has been removed, based on the additional references that are discussed now in the introduction.

AFM Setup

Comment Reviewer: It would be nice to see a schematic of the apparatus in the supplementary material.

We agree with the reviewer and have added a detailed description to the SI and a figure (FIG. S1).

Three load regimes

Comment Reviewer: How can these values be compared with existing rock physics experiments in order to better understand these processes at a larger scale? In particular, because the shear velocity explored in this study are very similar to the common velocities used in rock deformation experiments.

You are right. This is an important point that we had not properly explained in the manuscript. The single-asperity contact investigated here represents a single asperity in the field (e.g. fault gouge or rock roughness). While the loads at which these transitions happen will depend on the size of the asperity, the pressures required for the reported transitions can be directly compared across length scales. We have added the range of pressures for the three different mechanisms that are responsible for the frictional strength of this single-asperity along the manuscript. For instance, after the experimental results have been described and the different regimes identified, we summarize as follows:

Line 294: “In summary, by measuring the friction as a function of load and sliding velocity with an AFM tip sliding along a calcite surface in calcium chloride brines of different compositions, we resolved three different mechanisms responsible for the frictional strength of this nanoscale single asperity (FIG. 5a): hydration lubrication at normal stresses ($\sigma_n < 200$ MPa), in which the confined brine lubricates the nanoscale contact and the frictional strength is dictated by the viscous shear force of the fluid film; velocity-strengthening friction at higher normal stresses ($\sigma_n < 400$ MPa), as a result of a thermally activated slip of the surface-localized hydrated ions upon shear, strongly dependent on the solution composition; and pressure-solution facilitated slip which re-activates hydration lubrication at higher stresses ($\sigma_n > 400$ MPa) and sufficiently small sliding velocities and causes a significant weakening of the frictional strength. The latter was more pronounced with increasing calcite surface reactivity, which was achieved by increasing the calcium concentration in this work; in nature, it will depend on the specific composition of the brine.”

We also emphasize in the text that this is the stress at the level of a single asperity, which differs from the confining pressures in e.g. fault gouge experiments:

Line 113: “Importantly, this range of normal stresses is relevant to that along faults in the upper Earth crust, which is in the order of 100 MPa²⁵, but higher at single-asperity contacts, which determine the frictional

strength of faults. The confinement geometry provided by the AFM allows examining the slip phenomena at the scale of a single asperity.”

Page 3

FIGs. 1a-e show the friction force as a function of velocity at applied loads of 0.5 nN, 1 nN, and 2 nN (<200 MPa) in the LL-regime and for the selected concentrations of the brine.

Comment Reviewer: There is no description either here or in the supplementary material on how these tests were performed, making difficult to interpret the data. For example, how velocity was controlled and how was increased, stepwise?

Yes, stepwise. We have added this description to the SI.

“The AFM tip is driven by an internal piezo motor so that it slides 100 nm to the right and then to the left, thereby defining a friction loop, along the calcite surface at the selected constant velocity. The reflected laser by the cantilever shows the vertical and the lateral deflection of the cantilever in a photodetector, and the force is determined with the corresponding normal and lateral spring constants (k_{vert} and k_{lat}). When the AFM tip slides along the energy landscape of the calcite surface, it experiences a lateral force F_{lat} , which leads to a torsion of the cantilever. During friction force measurements, the applied load is maintained constant by a feedback loop, while the lateral force is determined with the measured lateral deflection of the cantilever and the lateral spring constant.

Isothermal friction force measurements at 25°C were performed on an atomically flat calcite surface along the (10 $\bar{1}$ 4) plane as a function of applied load (L) and velocity (v) at each selected concentration. The experimental set up and mechanic model are described in FIG. S1. The load-dependent friction force measurements were performed at sliding velocities of 0.2 and 2 $\mu\text{m/s}$ over a sliding distance of 100 nm, in which the sliding velocity was maintained constant by a piezo motor. At least ten friction loops were measured at each load to determine the average friction force and the standard deviation. Here, the load was ramped up (stepwise) 30 seconds after the ten friction loops were collected at each load. The velocity-dependent friction force measurements were conducted while the applied load was maintained constant; following loads were selected: 0.2, 0.5, 1, 2, 5, 10, 20, 30, 40 and 50 nN. The sliding velocity was increased stepwise from 0.1 to 10 $\mu\text{m/s}$ at each constant load, and also here, ten friction loops were measured at each velocity (per load) to determine the average friction force and the standard deviation. The load- and velocity-dependent friction measurements were repeated on three different spots on each calcite sample to confirm reproducibility.”

FIG. 1

Comment Reviewer: What is the orange and red curve in these panels? Those colors are not indicated in the legend. I guess they represent the data showed on the top panel but it should be mentioned somewhere.

The caption has been corrected. It says:

Line 112: “The color legend for the load is: orange (0.5 nN), red (1 nN), green (2 nN), turquoise (5 nN), purple (10 nN), blue (15 nN) and black (20 nN).”

Compare fig F and J are they the same friction trends??

There is a deviation from eq. (1) in j) at high load because pressure solution is already relevant. This is indicated in the text:

Line 247: “Although one cannot rule out the possibility that pressure solution also takes place in the IL-regime, the relation $F_L \sim \log(v)$ (FIG. 1) and the increasing F_L with load (FIG. 3) in this regime indicate that the influence of pressure solution is small.”

Page 4

F_L

Comment Reviewer: F_L is introduced here but never defined. From Gnecco et al., 2000 I can see that it is the Lateral Force. Is this correct?

Yes, it is the friction force obtained from lateral force measurements. The definition has been added to the manuscript:

Line 137: “This yields the following expression for the friction force F_L :”

FIG. 2a

Comment Reviewer: I would expect some more systematic changes of lambda as a function of concentrations

Yes, it was intriguing to us, as well. In fact, we observed very systematic changes in the case for NaCl on mica. We attribute the lack of systematic changes for calcite in CaCl_2 solutions to the overlap of a second mechanism, charge reversal, that happens at concentrations between 10 mM and 100 mM, according to our prior work¹. Because of the change of the charge of the interface, the interfacial composition changes and we lose the systematic trend. We explain this phenomenon in the manuscript:

Line 183: “As a matter of fact, the increasing friction force at concentrations above 10 mM coincides with the charge reversal of the confined calcite surface¹. We thus speculate that charge reversal of calcite magnifies the energy barrier above 10 mM via an additional electrostatic attraction to the naturally oxidized silicon tip, which is negatively charged. Consequently, at the same velocity, the greater friction force at concentrations above 10 mM can be justified. Although the conditions for charge reversal are mineral-specific, this phenomenon is expected to be universal for mineral surfaces in contact with calcium-rich brines.”

Page 5

F^* reflects the corrugation of the tip-substrate interaction potential, E_6 , and E_6 , and consequently F^* , were found to linearly increase with normal load.

Comment Reviewer: This sentence is not clear to me. It should explain the relation between F^* and E_0 ?

We have significantly simplified this description to make it more understandable:

Line 130: “The friction force follows an approximately linear relationship with the logarithm of velocity, i.e. $F_L \sim \log(v)$, thereby indicating a pathway of energy dissipation (friction) that differs from the viscous dissipation at smaller normal stresses. The logarithmic dependence of the friction force is attributed to a shear-promoted thermally activated slip in the context of transition state theory²⁷. Based on our previous work¹, we assume that (hydrated) calcium ions remain localized on oppositely charged surface sites under high applied pressures. Upon shear, the net frequency at which the hydrated ions can overcome the activation energy (ΔE) to crossing to an adjacent energy minimum is increased by the applied lateral force⁵.”

Line 147: “The activation energy ΔE of the thermally activated slip in a single-asperity contact under dry conditions increases linearly with the applied normal load⁴¹; intuitively, the same can be assumed to happen in a lubricated contact. The increase in the energy barrier with the increase in load thus makes sliding to be more discontinuous; the observed stick-slip confirms this (FIG. S4).”

Page 6:

This either negative (a) or smaller (b) F_L vs. L -slope indicates that slip is facilitated at the highest applied loads, but more significantly at slower sliding.

Comment Reviewer: This is not necessarily true since only for the case of $v = 0.2 \mu\text{m/s}$ there is a weakening in friction, otherwise friction is equal or higher. So, I think it is only the combination of high-load and low-velocity to induce weakening. This was also documented in Carpenter et al., 2016.

Very respectfully, note that the sentence talks about the slope of F_L vs. load and not about friction and that this slope decreases above a load of ~ 20 nN at the velocity of $2 \mu\text{m/s}$. Therefore, we think that the sentence is correct but unclear. Therefore, to avoid any misunderstanding, we say now:

Line 201: “For instance, while the sliding velocity is maintained constant at $0.2 \mu\text{m/s}$, friction decreases with increasing load above a critical value (L_c in FIG. 3a). A scrutiny of the friction force as a function of load at a full spectrum of sliding velocities (FIG. S8) shows that the decrease in the F_L vs. L -slope is facilitated at the highest applied loads, but more significantly at slower sliding velocities (e.g. see FIG. 3b at 2

$\mu\text{m/s}$). It is important to note that, when analogous friction measurements are conducted in a *dry* nitrogen atmosphere, i.e. in the absence of the brine (FIG. S9b), the friction force increases monotonically with applied load at all sliding velocities, which emphasizes the key role of the solution in decreasing the frictional strength of the single-asperity contact.”

Page 7

FIG.4

Comment Reviewer: Most of the curves looks to be S-shaped even in a log-log space. Why don't you show them in a linear-log space as in the inset?

As you can see below the log-log plots show better the differences between the friction forces measured at different loads and at small velocities. Therefore, we have chosen the log-log representation.

Comparison of log-linear and log-log scale for velocity-dependent friction force plot.

Comment Reviewer: cannot it be simply time-dependent contact growth induced by fluids and higher load? Or some plastic/dislocation processes? If this is the only evidence for pressure solution it is rather weak. How long were the experiments? Since it is not specified, but I can infer you slide for 100nm at 0.2 $\mu\text{m/s}$...
Comment Reviewer: These evidences are indicative but not directly related to pressure solution. I would like to see some images of the grain at the end of experiment to be fully certain.

Given the relevance of this aspect, we have performed additional experiments to demonstrate the phenomenon of pressure solution. We conducted friction force measurements in air, ethanol and brine and imaged the track of the tip on the substrate after the friction measurements. We describe the results as follows:

Line 222: “At the sliding conditions at which the F_L vs. L -slope was observed to decrease, imaging of the calcite surface showed the track of the tip (FIG. S11), while re-precipitation sometimes occurred close to the track (see the bright spot in FIG. S11c). Images taken after similar friction measurements in ethanol, in which calcite is insoluble, do not show any change of the surface, which excludes that brittle fracture and plastic deformation of calcite could be responsible for the observed track in the presence of the brine (FIG. S12). Further, we note that a similar decrease of friction with load was not detected for silica-silica contacts in aqueous solution (FIG. S7) and AFM imaging did not show any change of the silica surface and of the tip. Hence, the remarkable drop in friction with increase in load (FIG. 3) is attributed to the activation of the pressure solution of calcite at sufficiently high normal stresses and small sliding velocities.”

Please, see the figures in the SI, where we also explain:

“AFM images of calcite surfaces equilibrated in 1 mM $\text{CaCl}_2/\text{CaCO}_3$ before and after prolonged sliding at ~ 500 MPa showing the track of the AFM tip as a result of pressure solution of calcite. We note that after short sliding, like in the measurements shown in the main text, the track is not clearly visible (pressure applied for a time ≤ 20 s), and therefore, we prolonged here the sliding time to clearly demonstrate this

phenomenon. The dashed lines in Figures a-c) indicate where the sliding was conducted. After 1-min sliding, a track was detected. And a longer track was observed after sliding for another 1 min over a longer distance, while the topography changed nearby, which is attributed to the re-precipitation of dissolved calcite.”

We calculated the volume of the track and were able to roughly estimate a dissolution rate of $1.5 \cdot 10^{-10}$ mol/cm²/s, which is in the same order of magnitude as the dissolution rate of stress-free calcite surface. It is reasonable to expect that dissolved ions will re-precipitate back into the track concurrently with the sliding of the tip, which would justify an underestimation of the dissolution rate.

Page 10

It should be noted that the sliding velocities at which the pressure-solution facilitated sliding was observed in this work are in good agreement with the slip rates in slow-earthquake zones such as in San Andreas and Nankai trough (5-15 m/year).

Comment Reviewer: However, the rocks that characterize these plate boundary faults are totally different from pure calcite. In fact, in both the mentioned plate boundary fault phyllosilicates are the most abundant mineralogical phase present.

You are right. We have removed the comparison to slow earthquakes and focus the discussion the implications of our work regarding the frictional strength of saturated sedimentary rocks at the scale of a single asperity and the mechanisms of weakening. Please, see the modified discussion.

Manuscript: For instance, when two fault planes undergo creep slow enough for pressure solution to come into play, as the normal pressure surpasses a critical value, a slip can be triggered more favorably and alter the fate of tectonic activities.

Reviewer's Comment Reviewer: Please see the definition of fault plane, if you say two fault planes it means that you are considering two different faults. A fault plane is characterized by a discontinuity in the crust along which shear is localized. There are two fault surfaces and one plane.

Thank you. This has been corrected.

Line 370: “Hence, when a fault undergoes creep slow enough for pressure solution to come into play, a slip can be triggered more favorably and alter fault dynamics..”

Comment Reviewer: I don't think that it is so simple since seismogenic faults are greatly influenced by Temperature and many other factors.

We agree that it will be more complex in nature. Our conclusion says now:

Line 367: “In conclusion, we have identified the conditions that favor different friction mechanisms upon shear of a calcite-silica nanoscale asperity in the presence of a brine. We have demonstrated that pressure solution of calcite can directly impair the frictional strength of single asperity contacts providing hydration lubrication, similar to that occurring at much lower contact stresses. Hence, when a fault undergoes creep slow enough for pressure solution to come into play, a slip can be triggered more favorably and alter fault dynamics. This work shows that this stress must be larger than ~400 MPa for calcite asperities; similar phenomena are expected for other minerals but at different stresses. We also emphasize that the influence of the brine composition on mineral reactivity should be also considered, given the results of this study. This nanoscale evidence can be extended to fault weakening, although in this case, the relative contributions of subcritical crack growth, semi-brittle creep, and pressure-solution to macroscopic friction still remain to be quantified. The range of applied pressure and sliding velocities investigated in this study is of geological relevance to earthquake nucleation, while the additional effect of Earth's crust temperature on the proposed mechanisms calls for future studies.”

Reviewer #3 (Remarks to the Author):

This is an extremely thorough and thoughtful paper and provides nanoscale insight into geological processes. I recommend publication. The observations are novel and the conclusions appear justified. While it

may appear at first sight that nanoscale measurements with AFM are inappropriate to describe geological events, the local pressures are in fact relevant and comparable. The data is of extremely high quality and has been analysed with care. In terms of relevance, it is a joy to see fundamental physical chemistry being rigorously applied to explain geological phenomena.

The paper is clear and well written.

While the pressures and compositions are relevant, temperature is not mentioned. It is not necessary to show temperature dependent data, it behoves the authors to address the relevance of the temperatures of their experiments in the context of the seismic processes they purport to model.

Thank you very much for the positive evaluation of our manuscript and also for this important suggestion. We have learned that seismic ruptures typically nucleate in depths ranging between 5 km-15 km (see ref. ⁴²), where temperature ranges from 100 to 300°C, and hence, it is much higher than in our experiments. The frictional strength of fault gouges has been shown to depend on temperature (e.g. it depends for calcitic gouges but not for quartzitic gouges³⁹), and its temperature-dependence can be quite complex. Nevertheless, it is essential to understand the frictional response of crustal rocks at all depths (and temperatures) at sub-seismic sliding velocities to assess the probability that these rocks might contribute to earthquake rupture propagation, which is rather the objective of this work.

We have added following statement to the manuscript:

Line 374: “This nanoscale evidence can be extended to fault weakening, although in this case, the relative contributions of subcritical crack growth, semi-brittle creep, and pressure-solution to macroscopic friction still remain to be quantified. The range of applied pressure and sliding velocities investigated in this study is of geological relevance to earthquake nucleation, while the additional effect of Earth’s crust temperature on the proposed mechanisms calls for future studies.”

Minor Comment Reviewer

The statement that calcium does not populate mica needs perhaps to be justified: there are experimental studies that strongly imply population. (this is not central to the paper but is relevant to the general credibility)

Yes, you are right. It was not a precise statement. Our purpose was to emphasize that the surface forces measured by AFM do not reveal the presence of multiple layers of calcium and water at concentrations below 100mM, in contrast to our observations in the presence of e.g. Na⁺ ions. We agree that calcium ions strongly populate the mica surface, which is reflected in charge reversal at sufficiently high concentrations⁴³ and an oscillatory hydration force between mica surfaces at concentrations above 100 mM⁴⁴. We have corrected this statement as follows:

Line 164: “CaCl₂ was not selected for these experiments because it had been previously shown, and confirmed in our own measurements, that, although calcium ions adsorb on mica⁴³, the hydration force between mica surfaces in CaCl₂ solutions does not reveal the presence of multiple layers of ions and water at concentrations below 100 mM^{3,44,45}. In contrast, hydrated sodium ions accumulate on mica surfaces and form multiple layers at lower concentrations, similar to our observations for calcium ions on calcite¹.”

The use of the word “atop” on page 8 seems to be a typo. It might be politic to find an alternative expression to “humping velocity” on page 4.

Thank you, we have modified this explanation as follows:

Line 130: “The friction force follows an approximately linear relationship with the logarithm of velocity, i.e. $F_L \sim \log(v)$, thereby indicating a pathway of energy dissipation (friction) that differs from the viscous dissipation at smaller normal stresses. The logarithmic dependence of the friction force is attributed to a shear-promoted thermally activated slip in the context of transition state theory²⁷. Based on our previous work¹, we assume that (hydrated) calcium ions remain localized on oppositely charged surface sites under high applied pressures. Upon shear, the net frequency at which the hydrated ions can overcome the activation energy (ΔE) to crossing to an adjacent energy minimum is increased by the applied lateral force⁵.”

Thank you.
Mark Rutland

References

- 1 Diao, Y. J. & Espinosa-Marzal, R. M. Molecular insight into the nanoconfined calcite-solution interface. *Proceedings of the National Academy of Sciences of the United States of America* **113**, 12047-12052, doi:10.1073/pnas.1605920113 (2016).
- 2 Israelachvili, J. N. & Pashley, R. M. Molecular Layering of Water at Surfaces and Origin of Repulsive Hydration Forces. *Nature* **306**, 249-250 (1983).
- 3 Alcantar, N., Israelachvili, J. & Boles, J. Forces and ionic transport between mica surfaces: Implications for pressure solution. *Geochimica Et Cosmochimica Acta* **67**, 1289-1304, doi:Doi 10.1016/S0016-7037(02)01270-X (2003).
- 4 Røyne, A., Dalby, K. N. & Hassenkam, T. Repulsive hydration forces between calcite surfaces and their effect on the brittle strength of calcite-bearing rocks. *Geophysical Research Letters* **42**, 4786-4794 (2015).
- 5 Ma, L., Gaisinskaya-Kipnis, A., Kampf, N. & Klein, J. Origins of hydration lubrication. *Nature Communications* **6**, 6060 (2015).
- 6 Brace, W. F. & Byerlee, J. D. Stick-Slip as a Mechanism for Earthquakes. *Science* **153**, 990-&, doi:DOI 10.1126/science.153.3739.990 (1966).
- 7 Carpenter, B. M., Scuderi, M. M., Collettini, C. & Marone, C. Frictional heterogeneities on carbonate-bearing normal faults: Insights from the Monte Maggio Fault, Italy. *Journal of Geophysical Research: Solid Earth* **119**, 9062-9076, doi:10.1002/2014JB011337 (2014).
- 8 Carpenter, B. M., Collettini, C., Viti, C. & Cavallo, A. The influence of normal stress and sliding velocity on the frictional behaviour of calcite at room temperature: insights from laboratory experiments and microstructural observations. *Geophysical Journal International* **205**, 548-561, doi:10.1093/gji/ggw038 (2016).
- 9 Tesei, T. *et al.* Friction and scale-dependent deformation processes of large experimental carbonate faults. *Journal of Structural Geology* **100**, 12-23, doi:<https://doi.org/10.1016/j.jsg.2017.05.008> (2017).
- 10 den Hartog, S. A. M., Peach, C. J., de Winter, D. A. M., Spiers, C. J. & Shimamoto, T. Frictional properties of megathrust fault gouges at low sliding velocities: New data on effects of normal stress and temperature. *Journal of Structural Geology* **38**, 156-171, doi:<https://doi.org/10.1016/j.jsg.2011.12.001> (2012).
- 11 Dieterich, J. H. & Kilgore, B. D. Direct Observation of Frictional Contacts - New Insights for State-Dependent Properties. *Pure and Applied Geophysics* **143**, 283-302, doi:Doi 10.1007/Bf00874332 (1994).
- 12 Li, Q., Tullis, T. E., Goldsby, D. & Carpick, R. W. Frictional ageing from interfacial bonding and the origins of rate and state friction. *Nature* **480**, 233-236 (2011).
- 13 Tian, K. W. *et al.* Load and Time Dependence of Interfacial Chemical Bond-Induced Friction at the Nanoscale. *Physical Review Letters* **118**, 076103, doi:ARTN 076103 10.1103/PhysRevLett.118.076103 (2017).
- 14 Marone, C. Laboratory-derived friction laws and their application to seismic faulting. *Annu Rev Earth Pl Sc* **26**, 643-696, doi:DOI 10.1146/annurev.earth.26.1.643 (1998).
- 15 Ruina, A. Slip Instability and State Variable Friction Laws. *Journal of Geophysical Research* **88**, 359-370, doi:DOI 10.1029/JB088iB12p10359 (1983).

- 16 Persson, B. N. J. *et al.* On the nature of the static friction, kinetic friction and creep. *Wear* **254**, 835-851, doi:10.1016/S0043-1648(03)00234-5 (2003).
- 17 Scuderi, M. M. & Collettini, C. The role of fluid pressure in induced vs. triggered seismicity: insights from rock deformation experiments on carbonates. *Scientific Reports* **6**, 24852, doi:10.1038/srep24852
<https://www.nature.com/articles/srep24852-supplementary-information> (2016).
- 18 Scuderi, M. M., Collettini, C. & Marone, C. Frictional stability and earthquake triggering during fluid pressure stimulation of an experimental fault. *Earth and Planetary Science Letters* **477**, 84-96, doi:<https://doi.org/10.1016/j.epsl.2017.08.009> (2017).
- 19 Iler, R. K. *The Chemistry of Silica: Solubility, Polymerization, Colloid and Surface Properties and Biochemistry of Silica.* (Wiley, 1979).
- 20 Chen, J., Ratera, I., Park, J. Y. & Salmeron, M. Velocity Dependence of Friction and Hydrogen Bonding Effects. *Physical Review Letters* **96**, 236102 (2006).
- 21 Liu, J., Notbohm, J. K., Carpick, R. W. & Turner, K. T. Method for characterizing nanoscale wear of atomic force microscope tips. *ACS nano* **4**, 3763-3772 (2010).
- 22 Gao, J. *et al.* Frictional Forces and Amontons' Law: From the Molecular to the Macroscopic Scale. *The Journal of Physical Chemistry B* **108**, 3410-3425, doi:10.1021/jp036362l (2004).
- 23 Israelachvili, J. N. *Intermolecular and surface forces: revised third edition.* (Academic press, 2011).
- 24 Wilson, B., Dewers, T., Reches, Z. e. & Brune, J. Particle size and energetics of gouge from earthquake rupture zones. *Nature* **434**, 749, doi:10.1038/nature03433 (2005).
- 25 Sibson, R. H. Power Dissipation and Stress Levels on Faults in the Upper Crust. *Journal of Geophysical Research* **85**, 6239-6247, doi:DOI 10.1029/JB085iB11p06239 (1980).
- 26 Chhabra, R. P. in *Rheology of complex fluids* 3-34 (Springer, 2010).
- 27 Eyring, H. The Activated Complex in Chemical Reactions. *The Journal of Chemical Physics* **3**, 107-115, doi:10.1063/1.1749604 (1935).
- 28 Schott, J. *et al.* Dissolution kinetics of strained calcite. *Geochimica et Cosmochimica Acta* **53**, 373-382, doi:[https://doi.org/10.1016/0016-7037\(89\)90389-X](https://doi.org/10.1016/0016-7037(89)90389-X) (1989).
- 29 Helgeson, H. C., Murphy, W. M. & Aagaard, P. Thermodynamic and kinetic constraints on reaction rates among minerals and aqueous solutions. II. Rate constants, effective surface area, and the hydrolysis of feldspar. *Geochimica et Cosmochimica Acta* **48**, 2405-2432, doi:[https://doi.org/10.1016/0016-7037\(84\)90294-1](https://doi.org/10.1016/0016-7037(84)90294-1) (1984).
- 30 Greene, G. W., Kristiansen, K., Meyer, E. E., Boles, J. R. & Israelachvili, J. N. Role of electrochemical reactions in pressure solution. *Geochimica et Cosmochimica Acta* **73**, 2862-2874, doi:<https://doi.org/10.1016/j.gca.2009.02.012> (2009).
- 31 Cubillas, P., Kohler, S., Prieto, M., Chairat, C. & Oelkers, E. H. Experimental determination of the dissolution rates of calcite, aragonite, and bivalves. *Chemical Geology* **216**, 59-77, doi:10.1016/j.chemgeo.2004.11.009 (2005).
- 32 Audet, P. & Schwartz, S. Y. Hydrologic control of forearc strength and seismicity in the Costa Rican subduction zone. *Nature Geoscience* **6**, 852-855 (2013).
- 33 Violay, M. *et al.* Effect of water on the frictional behavior of cohesive rocks during earthquakes. *Geology* **42**, 27-30 (2014).
- 34 Violay, M. *et al.* Pore fluid in experimental calcite-bearing faults: Abrupt weakening and geochemical signature of co-seismic processes. *Earth and Planetary Science Letters* **361**, 74-84, doi:<https://doi.org/10.1016/j.epsl.2012.11.021> (2013).
- 35 Masuda, K., Arai, T., Fujimoto, K., Takahashi, M. & Shigematsu, N. Effect of water on weakening preceding rupture of laboratory-scale faults: Implications for long-term weakening of crustal faults. *Geophysical Research Letters* **39**, doi:Artn L01307
 10.1029/2011gl050493 (2012).

- 36 Griggs, D. Hydrolytic Weakening of Quartz and Other Silicates. *Geophys J Roy Astr S* **14**, 19-31 (1967).
- 37 Griggs, D. T., Blacic, J. D., Christie, J. M., McLaren, A. C. & Frank, F. C. Hydrolytic Weakening of Quartz Crystals. *Science* **152**, 674-&, doi:DOI 10.1126/science.152.3722.674-a (1966).
- 38 Royne, A., Bisschop, J. & Dysthe, D. K. Experimental investigation of surface energy and subcritical crack growth in calcite. *J Geophys Res-Sol Ea* **116**, doi:Artn B04204 10.1029/2010jb008033 (2011).
- 39 Verberne, B. A., He, C. R. & Spiers, C. J. Frictional Properties of Sedimentary Rocks and Natural Fault Gouge from the Longmen Shan Fault Zone, Sichuan, China. *B Seismol Soc Am* **100**, 2767-2790, doi:10.1785/0120090287 (2010).
- 40 Verberne, B. A. *et al.* Frictional Properties and Microstructure of Calcite-Rich Fault Gouges Sheared at Sub-Seismic Sliding Velocities. *Pure and Applied Geophysics* **171**, 2617-2640, doi:10.1007/s00024-013-0760-0 (2014).
- 41 Riedo, E., Gnecco, E., Bennewitz, R., Meyer, E. & Brune, H. Interaction potential and hopping dynamics governing sliding friction. *Phys Rev Lett* **91**, 084502, doi:10.1103/PhysRevLett.91.084502 (2003).
- 42 Di Toro, G. *et al.* Fault lubrication during earthquakes. *Nature* **471**, 494-498 (2011).
- 43 Kékicheff, P., Marčelja, S., Senden, T. J. & Shubin, V. E. Charge reversal seen in electrical double layer interaction of surfaces immersed in 2:1 calcium electrolyte. *The Journal of Chemical Physics* **99**, 6098-6113, doi:10.1063/1.465906 (1993).
- 44 Kjellander, R., Marcelja, S., Pashley, R. M. & Quirk, J. P. A theoretical and experimental study of forces between charged mica surfaces in aqueous CaCl₂ solutions. *The Journal of Chemical Physics* **92**, 4399-4407, doi:10.1063/1.457750 (1990).
- 45 Parsons, D. F. & Ninham, B. W. Charge reversal of surfaces in divalent electrolytes: the role of ionic dispersion interactions. *Langmuir* **26**, 6430-6436 (2010).

Reviewers' comments:

Reviewer #2 (Remarks to the Author):

Dear Editor

I have now reviewed for the second time the article entitled "About how water lubricates faults" by Diao Y. and Espinoza-Marzal R. The authors have performed a great amount of work from the first submission that greatly improved the manuscript quality and clarity. The manuscript now reads smoothly and fortunately they added a discussion to their data. They also have better contextualized their work at the light of the previous literature. They have properly answered all my previous concerns. For these reasons I recommend for publication and some minor revisions as reported in the line-by-line below.

Sincerely.

L 9: "The friction between two adjacent tectonic plates undergoing shear may dictate seismic activities" this sentence is not clear to me, please rephrase.

L10: "understanding the mechanisms"

L 26: I would define RSF "constitutive equations" since they are empirically derived and the physics of the processes that they describe is still poorly understood.

L 27: "tectonic fault zone processes" This is not very correct since fault zones undergo a variety of processes that may include hydromechanical processes and RSF do not describe them. I would rephrase deleting processes.

L 38-39: Usually, fault zones, in particular during the span of a seismic cycles, slide in one direction as applied remotely by plates movements. There are cases where a faults, let's say a thrust fault, is reactivated as normal fault but this change in deformation and stress field do not take place during one seismic cycle. So, I do not understand how this statement is relevant to tectonic faults.

L39-40: I think this is overstated.

L 41: "Water may play a significant role on earthquake ruptures" It's a very general statement that do not connect with the end of the sentence since you have not described any phenomenon. Earthquake rupture is generally divided in nucleation and propagation and water activates different mechanisms depending on the stage of faulting (e.g. lubrication by thermal pressurization during dynamic propagation). Since here you are investigating slow sliding rates I would suggest rewording as " Water plays an important role in the characterization of earthquake nucleation" or something on the line.

L 43: please delete "in subduction zones"

L 43: please change seismic with seismologic

L 44: "Frictionally unfavorable" What does it means? frictionally strong or unfavorable oriented to the stress field? Looking at the references I think you are referring to a velocity strengthening fault.

L 53: change strength with behavior

L 88: I would substitute "in the frictional response" with "based on the frictional response to sliding"

L 100: is it shear rate or shear strain rate? Since I see a velocity divided by a thickness I suppose that is shear strain rate

L 117: In this equation do you assume that D (the film fluid thickness is constant)?

L 120: are you referring to the n exponent?

L 126: "high normal stresses" Specify that relative to that study and not in the current since you are in the LL regime.

L 140: why V_0 is a reference attempted?

L 205: I would change with "compare fig 3a and b" since you are highlighting the weakening induced by velocity in this sentence.

L 231: I think that a velocity is slow not small.

L 290: "high normal stress .." and slow slip velocity, which to me is the most relevant aspect.

Reviewer #3 (Remarks to the Author):

The new version of the manuscript appears (to this untutored eye) to be more rigorous towards the geological aspects - the physical chemistry and nanotribology was already good. I would have liked to see the expected temperature effects to be addressed a bit more fully but the authors are clearly being pulled in different directions and it is probably unreasonable for a paper of this length.

I reiterate my original opinion that it should be published.

Reviewer #4 (Remarks to the Author):

Review of "About how water lubricates faults" by Yijue Diao, Rosa. M. Espinosa-Marzal

In this manuscript the authors present new and interesting data on frictional behavior of single calcite asperity immersed in aqueous solutions and sheared at low slip velocity. Recent years have seen many publications on the frictional properties of calcite-bearing rocks, which all show that friction is governed by micromechanical mechanisms operating at the asperity scale. However, the experiments were often performed on cm-size samples. In this sense, it is refreshing to read a study such as this one, in which friction is measured by atomic force microscopy on nm-size samples. The main results presented by the authors are that calcite asperities are lubricated by 3 distinct mechanisms depending of the applied stress: viscous shear regime (at low stress), thermally activated slip regime (at intermediate stress), and pressure solution facilitated slip regime (at high stress). The number of experiments shown in this paper represents an impressive volume of work. However, the paper is poorly written, not concise and not very well organized that it is difficult to understand what the authors are saying. I recommend publication after the following major comments have been suitably considered.

Main comments:

1) My first comment concerns the structure and organization of the paper.

- The work merits publication, but as it stands, it could be improved by making the results more accessible to the general readership. In its current form it reads more as a very specialized paper: dealing with details and specifics - the authors could address this by trying to more simply summarize the results. For more clarity, I also recommend to the authors to be more cautious in distinguishing between results and interpretation (discussion).

- I am uneasy about the amount of material that sits within the Supplementary material sections. Ideally it would be good to see more of this material in the main body of the paper but that of course makes the paper too long. Especially it would be good to present the data of the dry experiment (FIG S9) within the main text. As you revise the paper, try to trim out other parts of the current text and make room for some of the material that sits in the Supplementary sections. I realize that there are limits to what you can do here, but still feel it is worth asking you to do this.

- I do find the conclusions difficult to fully extract and appreciate. The authors need to "come up to surface" and more clearly articulate the worth of their work.

2) My second comment concerns the introduction. It is not clear to me what mechanisms the authors are trying to understand. For instance (L41-56), I do not understand if they want to discuss the effect of water on earthquake nucleation (low slip rate, Scuderi et al., 2017 for example), or the effect of water on fault lubrication during earthquake propagation (high slip rate, Violay et al., 2013, 2014) or (maybe) the effect of water on fault healing during inter-seismic period....

3) Based on the experimental difficulty of this study, the description of the experimental method in lines 57-115 is certainly disappointing. For instance, clear description of the experimental protocol is missing in the main text (even if partially present in sup mat). The corrections/processing applied to the measured force and displacement are absent. I suggest rewriting this paragraph and being more specific. Moreover, experimental curves in FIG S4 need to be better explained. I also note that the total displacement during a test is much smaller than the size of the tip radius. Could the authors discuss how this might impact their results?

4) My fourth point concerns the comparison between dry and wet experiments. From the beginning until the end of the paper, the authors argue that water is very important for fault weakening mechanism and earthquake propagation. That may be the case, but I see no direct evidence in the paper. I think the authors must present the experiment performed under dry condition (Fig s9) within the main text, and then detail the 3 deformation regimes under fluid conditions. Discussion section should start with a comparison between dry and wet experiments. Dry experiments need to be reported on fig.5.

5) The explanations of the viscous shear and thermally-activated regime are unclear. What is the effect of fluid viscosity in the LL regime? Does viscosity change with fluid salinity? Is the initial brine a Newtonian fluid? Moreover, the paragraph regarding the thermo-activated slip regime, on lines 130-138, is not written in a way that is accessible to a wide audience. Part of this is stylistic and could be addressed by improving the writing to make more fluid and idiomatic, but a second part of this involves the heavy use of jargon and very specialized language. Please, explain the "transition state theory". What is a net frequency? Adjacent energy minimum?

6) L326-380: The discussion on the implications of the authors' results to fault lubrication is probably the weakest point of the manuscript. First, I do not really understand the message that the authors want to convey. I believe they tried to compare nano-scale friction to cm-scale friction experiments. However, I have the impression that the relationship between these 2 measurements is not obvious. For instance, the friction coefficients measured in this study are ~ 0.01 whereas it is assumed that calcite bearing rocks have a friction coefficient around 0.6. Can the authors comment on this point?

Minor comment:

L 9: shear loading

L10: remove : "understanding.... relevance".

L22: fault irreversible deformation

L26: Rate and State Friction (RSF)

L27: RSF is used to understand earthquake nucleation, not propagation.

L28: granular material = gouge

L33-26: "aging of silica-silica... across the interface". Please avoid the use of jargon and very specialized language.

L 62 : "located in an acoustic chamber". Please avoid the use of jargon and very specialized language. Explain better the experimental protocol.

L93: I do not understand what 'confinement geometry' means.

Fig1: Figure 1 and figure 4 should be merged to a single figure that must show the 3 deformation regimes

Fig S9 should be within the main text and inserted before figure 1. This should help the reader understanding why the authors are talking about water lubrication.

L142-146: 3 different definitions of the coherence length are given. Please simplify and give only one definition.

Fig1 and Fig4: curves 10nN and 20nN are plotted in both fig1 and fig4. Please precise if 10nN and 20 nN are in the IL or in the HL regime.

Fig.5: please plot the dry experiments on fig. 5.

Fig.5: In calcite friction coefficient is assumed to be ~ 0.5 . Can the author explain the low ratio between the friction force and the friction load (~ 0.01) ?

Fig.5: please explain better the panels b, c, d

L 307-310: I am not really convinced that Fig, 5b shows discontinuous sliding. I see only noise.

Dear editor and reviewers,

Thank you very much again for all your comments. We are glad to see that this revision is considered to have improved significantly. We also appreciate Reviewer 4's comments, which have helped to strengthen this manuscript even further. We have addressed each comment point by point in this document and have highlighted the changes in the manuscript. The introduction has been modified and we give now a general background, followed by the gap in the knowledge as motivation for this work, and a short summary of the study. The manuscript has been also read by a colleague that is not involved in the manuscript and he has made several suggestions to improve the clarity of the manuscript, from the introduction to the discussion. We wish that you will find the resubmission of both the manuscript and the SI to be appropriate for publication. Thanks again for this opportunity.

Reviewer #2 (Remarks to the Author):

Dear Editor Dr. Lewis Collins,

I have now reviewed for the second time the article entitled "About how water lubricates faults" by Diao Y. and Espinoza-Marzal R. The authors have performed a great amount of work from the first submission that greatly improved the manuscript quality and clarity. The manuscript now reads smoothly and fortunately they added a discussion to their data. They also have better contextualized their work at the light of the previous literature. They have properly answered all my previous concerns. For these reasons I recommend for publication and some minor revisions as reported in the line-by-line below.

Sincerely,

Dear reviewer,

We sincerely thank you for your immense input. We are glad to know that you found it smooth and pleasant to read, and that we have successfully addressed all your previous concerns. We thank you for the line-by-line minor revisions you pointed out. We have addressed them accordingly.

L 9: "The friction between two adjacent tectonic plates undergoing shear may dictate seismic activities" this sentence is not clear to me, please rephrase.

We have revised this sentence to be:

In the abstract:

"The friction between two adjacent tectonic plates under shear loading may dictate seismic activities. To advance the understanding of the mechanisms underlying the frictional strength of fault rocks, we investigate the frictional characteristics of calcite in aqueous environment".

In the introduction:

"When the static friction between two adjacent tectonic plates under shear loading is overcome, the two plates move relative to each other, which is known as a fault slip; under conditions of unstable slip, that is, when an intermittent motion (or *stick-slip*) occurs, an earthquake rupture may take place¹."

L10: "understanding the mechanisms".

We thank the reviewer to point out the absence of the article. We have added "the" into the sentence.

L 26: I would define RSF "constitutive equations" since they are empirically derived and the physics of the processes that they describe is still poorly understood.

We thank the reviewer for this point. We agree that the empirically derived RSF is still poorly understood.

To emphasize this, we introduce them as:

"Rate and state friction (RSF) constitutive equations^{2,3} are commonly used to phenomenologically describe the frictional response of tectonic fault zones and earthquake nucleation."

And later in the manuscript, we refer to them as "RSF equations", for short.

L 27: “tectonic fault zone processes” This is not very correct since fault zones undergo a variety of processes that may include hydromechanical processes and RSF do not describe them. I would rephrase deleting processes.

Thank you for your suggestion. We have rephrased as you suggested.

L 38-39: Usually, fault zones, in particular during the span of a seismic cycles, slide in one direction as applied remotely by plates movements. There are cases where a fault, let's say a thrust fault, is reactivated as normal fault but this change in deformation and stress field do not take place during one seismic cycle. So, I do not understand how this statement is relevant to tectonic faults.

We apologize for this confusion. We meant that AFM studies indicated that the crystal anisotropy may lead to different frictional behaviors along different sliding directions, but not necessarily in one single seismic cycle. We have rephrased the sentence to:

“A precedent study of the frictional characteristics of calcite under dry conditions by AFM showed that both stick-slip behavior and kinetic friction depend on the sliding direction.”⁴

L39-40: I think this is overstated.

We modified it by changing the sentence to:

“While these recent works illustrate the first attempts to bridge the gap between nanoscale friction measurements and fault dynamics, many questions still remain open.”

L 41: “Water may play a significant role on earthquake ruptures” It's a very general statement that do not connect with the end of the sentence since you have not described any phenomenon. Earthquake rupture is generally divided in nucleation and propagation and water activates different mechanisms depending on the stage of faulting (e.g. lubrication by thermal pressurization during dynamic propagation). Since here you are investigating slow sliding rates I would suggest rewording as “ Water plays an important role in the characterization of earthquake nucleation” or something on the line.

Thank you very much. We have changed it to:

“Along these lines, it has been proposed that water may play an important role in fault dynamics and earthquake nucleation^{5,6}, which has motivated this work.

L 43: please delete “in subduction zones”

Thank you. We have deleted this.

L 43: please change seismic with seismologic

Thank you. We have changed it accordingly.

L 44: “Frictionally unfavorable” What does it means? frictionally strong or unfavorable oriented to the stress field? Looking at the references I think you are referring to a velocity strengthening fault. Yes, by “friction unfavorable”, we mean that the high pore fluid pressure allows earthquake slip to occur along the fault even it is velocity-strengthening, in which case unstable slip is not expected according to the RSF prediction. We changed the sentence to clarify this point:

“Regarding the effect of water, RSF equations predict that fluid overpressure decreases the critical fault stiffness, and therefore, it should promote aseismic creep. In contrast to these predictions, seismologic studies⁵ as well as laboratory experiments⁷ suggest that a high fluid pore pressure can trigger a dynamic slip instability, perhaps as a result of the decrease of the effective compressive stress, which brings the fault closer to failure.”

L 53: change strength with behavior

Thank you. We have modified it to:

“For instance, water may promote fracture of asperities by chemically reacting with the strained apexes of microcracks (also called subcritical crack growth)⁸, thereby lowering the resistance to slip, i.e. friction.”

L 88: I would substitute “in the frictional response” with “based on the frictional response to sliding”
Thank you. We have revised it accordingly.

“Based on the measured frictional behavior of calcite in aqueous solutions, three different mechanisms of energy dissipation were distinguished”

L 100: is it shear rate or shear strain rate? Since I see a velocity divided by a thickness I suppose that is shear strain rate

Thank you very much. You are right. It is shear strain rate and it has been changed accordingly.

L 117: In this equation do you assume that D (the film fluid thickness) is constant?

Yes, the underlying assumption is that D remains constant. D cannot be explicitly measured with an AFM since the absolute separation between the tip and the substrate is unknown. We have added following explanation:

“Assuming that the change in D is negligible across the investigated range of loads and velocities, the average exponent n for all selected concentrations is estimated to be 0.38 ± 0.06 . An exponent n smaller than 1 indicates that the confined fluid behaves as a non-Newtonian fluid with shear-thinning behavior, i.e. its viscosity decreases with increasing shear strain rates. A calculation of the viscosity is, however, not possible, because the absolute separation between the tip and the calcite surface (D) cannot be unambiguously determined. Nevertheless, a similar shear-thinning behavior has been observed for confined water between mica and an AFM tip at stresses ≥ 100 MPa⁹, while a constant viscosity was reported at much smaller normal stresses (< 1 MPa)¹⁰. This suggests that the high normal stresses applied in the LS regime may be responsible for the observed non-Newtonian behavior of the confined aqueous solution.”

L 120: are you referring to the n exponent?

Yes, and we have added “ n ” to clarify it.

L 126: “high normal stresses” Specify that relative to that study and not in the current since you are in the LL regime.

Thank you. We say now:

“This suggests that the high normal stresses applied in the LS regime may be responsible for the observed non-Newtonian behavior of the confined aqueous solution.”

L 140: why V_0 is a reference attempted?

v_0 describes the fluctuation rate of the molecules (water) and ions in a reference state. We have rephrased this.

“ k_B being the Boltzmann constant, T the absolute temperature, v_0 the molecular fluctuation rate at a reference state, and λ the stress-activation length.”

L 205: I would change with “compare fig 3a and b” since you are highlighting the weakening induced by velocity in this sentence.

Thank you. We have modified the sentence to:

“The decrease in the friction coefficient -i.e. the slope of F_L vs. L - is more significant at slower sliding velocities (compare FIG. 3a and 3b).”

L 231: I think that a velocity is slow not small.

Thank you very much. We have corrected it to “slow sliding velocities”.

L 290: “high normal stress” and slow slip velocity, which to me is the most relevant aspect.

Thank you. We have added “slow sliding velocity” to this sentence.

Reviewer #3 (Remarks to the Author):

The new version of the manuscript appears (to this untutored eye) to be more rigorous towards the geological aspects - the physical chemistry and nanotribology was already good. I would have liked to see the expected temperature effects to be addressed a bit more fully but the authors are clearly being pulled in different directions and it is probably unreasonable for a paper of this length. I reiterate my original opinion that it should be published.

Dear reviewer,

Thank you very much for your supportive evaluation as well as your input that help us to improve this manuscript. We are also glad to know that we have delivered the geological insight to our tribologist colleagues. We would also like to explore in more depth towards the effect of temperature. For instance, since the solubility of calcite is temperature-dependent, a friction study as a function of temperature will provide more insight into the mechanism of pressure-solution facilitated slip that we have proposed. However, given the word limit for this paper, this will go into our future work.

Reviewer #4 (Remarks to the Author):

Review of "About how water lubricates faults" by Yijue Diao, Rosa. M. Espinosa-Marzal.

In this manuscript the authors present new and interesting data on frictional behavior of single calcite asperity immersed in aqueous solutions and sheared at low slip velocity. Recent years have seen many publications on the frictional properties of calcite-bearing rocks, which all show that friction is governed by micromechanical mechanisms operating at the asperity scale. However, the experiments were often performed on cm-size samples. In this sense, it is refreshing to read a study such as this one, in which friction is measured by atomic force microscopy on nm-size samples. The main results presented by the authors are that calcite asperities are lubricated by 3 distinct mechanisms depending of the applied stress: viscous shear regime (at low stress), thermally activated slip regime (at intermediate stress), and pressure solution facilitated slip regime (at high stress). The number of experiments shown in this paper represents an impressive volume of work. However, the paper is poorly written, not concise and not very well organized that it is difficult to understand what the authors are saying. I recommend publication after the following major comments have been suitably considered.

Dear reviewer,

Thank you very much for your critical comments and your recommendation for publication. We have reviewed the manuscript according to your comments and corrections, which has helped improve our manuscript remarkably and we are very thankful. A colleague of us also read the manuscript so that we were able to explain some concepts more clearly. We hope you find all your concerns thoroughly considered and this revision is suitable for publication.

Main comments:

1) My first comment concerns the structure and organization of the paper. The work merits publication, but as it stands, it could be improved by making the results more accessible to the general readership. In its current form it reads more as a very specialized paper: dealing with details and specifics - the authors could address this by trying to more simply summarize the results.

For more clarity, I also recommend to the authors to be more cautious in distinguishing between results and interpretation (discussion).

- I do find the conclusions difficult to fully extract and appreciate. The authors need to "come up to surface" and more clearly articulate the worth of their work.

Thank you very much. We have reviewed the manuscript, paying attention to the structure and organization of the paper, as well as the distinction between results and discussion. The introduction has been also modified and we give a general background, followed by the gap in the knowledge as motivation

for this work, and a short summary of the study. We have also modified the discussion and conclusions and hope that they are clearer now.

- I am uneasy about the amount of material that sits within the Supplementary material sections. Ideally it would be good to see more of this material in the main body of the paper but that of course makes the paper too long. Especially it would be good to present the data of the dry experiment (FIG S9) within the main text. As you revise the paper, try to trim out other parts of the current text and make room for some of the material that sits in the Supplementary sections. I realize that there are limits to what you can do here, but still feel it is worth asking you to do this.

We really appreciate your suggestion. Admittedly, we have put a significant amount of supporting experiments in the SI. We agree that including the dry contact friction into the manuscript is helpful and we have included it in the revised manuscript as Fig. 4 and also in Fig. 5, as suggested. We found that it was smoother to start with wet friction and show later the dry friction in Fig. 4 (and 5). Moreover, we have removed the experimental methods from the SI, and have created a dedicated section for the Methods in the main manuscript. We have also included FIG. S2 in the insets of FIGs. 1. We have also removed FIG. S4 since the information was already included in FIGs. 5b-d. Unfortunately including the reference measurements with mica, the silicon nitride tip and the silica substrate into the main text would make the manuscript very long, and therefore they have been kept in the SI. We also liked very much your suggestion to merge the old FIG. 4 and FIG. 1 into one figure, and now Fig. 1 shows the velocity-dependent friction force in the three regimes. We changed the color scheme: the results in the LS regime are in blue, in the IS regime in green and in the HS regime in red to achieve more clarity.

2) My second comment concerns the introduction. It is not clear to me what mechanisms the authors are trying to understand. For instance (L41-56), I do not understand if they want to discuss the effect of water on earthquake nucleation (low slip rate, Scuderi et al., 2017 for example), or the effect of water on fault lubrication during earthquake propagation (high slip rate, Violay et al., 2013, 2014) or (maybe) the effect of water on fault healing during inter-seismic period....

We thank to the reviewer for this critical comment. Since we focus on slow slip rate in our experiments, we aim to relate our results to earthquake nucleation. We have modified the introduction accordingly and we hope you find it suitable now.

3) Based on the experimental difficulty of this study, the description of the experimental method in lines 57-115 is certainly disappointing. For instance, clear description of the experimental protocol is missing in the main text (even if partially present in sup mat). The corrections/processing applied to the measured force and displacement are absent. I suggest rewriting this paragraph and being more specific. Moreover, experimental curves in FIG S4 need to be better explained. I also note that the total displacement during a test is much smaller than the size of the tip radius. Could the authors discuss how this might impact their results?

Thank you very much for pointing this out. We have extended the description of the experimental method in a dedicated section at the end of the manuscript. The description of the friction loops can now be found in both the Methods section and in the main text. We hope the reviewer will find this revised version more accessible.

Concerning the comment about the total displacement in comparison with the tip radius. We note that what matters in this type of experiments is the contact radius which is smaller than 10 nm under all conditions. Nevertheless, we had also performed experiments using sliding distances of 200 nm and 1 μm , and they did not show any statistically significant difference; there is a very good agreement. We show the results for 100 nm because we have the most complete series of experiments and more repetitions (note that the experiments take longer with longer sliding distances). We explain the comparison of results measured with tips with different radii in the Methods section.

4) My fourth point concerns the comparison between dry and wet experiments. From the beginning until the end of the paper, the authors argue that water is very important for fault weakening mechanism and

earthquake propagation. That may be the case, but I see no direct evidence in the paper. I think the authors must present the experiment performed under dry condition (Fig s9) within the main text, and then detail the 3 deformation regimes under fluid conditions. Discussion section should start with a comparison between dry and wet experiments. Dry experiments need to be reported on fig.5.

We appreciate the reviewer's suggestion very much and we agree on the relevance of this comparison. Figure 4 and 5 have been now updated to include the dry friction experiment. Moreover Fig. 3 shows a comparison of the coefficients of friction obtained from load-dependent friction measurements at constant sliding velocity. The Discussion section now starts with a comparison of dry and wet friction coefficients as follows:

“Implications of our work regarding the frictional strength of calcite-rich faults are discussed next. Our experiments have showed that the presence of the aqueous solution leads to a significant weakening of the frictional strength of the single-asperity contact compared to dry conditions, as the friction coefficient drops from ~ 0.05 to lower than ~ 0.005 (FIG. 3a). Previous studies of calcite-rich fault gouge reported a decrease in the friction coefficient from ~ 0.7 to ~ 0.6 , i.e. much less pronounced¹¹. This is, however, not surprising, since granular flow, asperity interlocking and fracture as well as plastic deformation of multiple contacts can concurrently happen in both dry and wet experiments with (simulated) fault gouges and rock surfaces under shear loading, and they significantly increase friction¹². The simple geometry of our study enables us to exclude these complex phenomena, and hence, to evaluate the direct effect of hydration lubrication and pressure solution on the frictional characteristics of calcite. “

5) The explanations of the viscous shear and thermally-activated regime are unclear.

We agree with the reviewer and have developed these explanations further, as described next.

What is the effect of fluid viscosity in the LL regime? Does viscosity change with fluid salinity? Is the initial brine a Newtonian fluid?

The viscosity of the solution film increases slightly with solution concentration. Unfortunately, AFM does not allow one to measure the absolute separation between the tip and the surface, and therefore the film thickness D is unknown. This is the reason why we only give the friction as a function of the velocity (V) and not of the shear strain rate (V/D). Under these conditions, one cannot quantify the viscosity. We are currently developing a method to perform similar experiments, but with a surface forces apparatus, in which D will be directly measured.

The description has been modified as follows:

“FIGs. 1a-e show the friction force as a function of velocity at applied loads of 0.5 nN, 1 nN, and 2 nN ($\sigma_n \leq 200$ MPa) and at calcium chloride concentrations of 0 mM (no CaCl_2 added), 1 mM, 10 mM, 100 mM, and 1 M to simulate calcium concentrations in nature, from ground water¹³ to concentrated brines¹⁴. As shown here, friction increases with load but the change is very small in this narrow range of loads. The velocity-dependence of the friction force in the LS regime can be fit by a power law (see solid lines), which is reminiscent of the shear response of fluids as described by the Ostwald-de Waele equation¹⁵

$$F_L \sim A\eta_0(v/D)^n \quad \text{Eq. (1)}$$

F_L being the friction force, A the area, η_0 a constant, v/D the shear strain rate, D the fluid film thickness, and n an exponent. This relation suggests that friction in the LS-regime is mainly due to a viscous force that results from shearing the confined aqueous solution. The lubrication mechanism provided by a nanoconfined solution has been previously referred to as “hydration lubrication”¹⁰.

The composition of the confined solution between calcite and a (micron-sized) silica tip was scrutinized in our previous AFM study¹⁶, by measuring the normal surface forces down to subnanometer calcite-silica separations. This work showed that a strong repulsion –termed disjoining pressure or hydration force¹⁷⁻¹⁹ – is originated by the confinement of the solution between the two surfaces and it prevents the solution to be squeezed-out under an applied pressure. The confined solution has a thickness of a few nanometers and is composed of water layers that are strongly adsorbed to the calcite surface and located underneath a layer of partially dehydrated calcium ions, and fully hydrated calcium ions further away from the surface.

Assuming that the change in D is negligible across the investigated range of loads and velocities, the average exponent n for all selected concentrations is estimated to be 0.38 ± 0.06 . An exponent n smaller than 1 indicates that the confined fluid behaves as a non-Newtonian fluid with shear-thinning behavior, i.e. its viscosity decreases with increasing shear strain rates. A calculation of the viscosity is, however, not possible, because the absolute separation between the tip and the calcite surface (D) cannot be unambiguously determined. Nevertheless, a similar shear-thinning behavior has been observed for confined water between mica and an AFM tip at stresses ≥ 100 MPa⁹, while a constant viscosity was reported at much smaller normal stresses (< 1 MPa)¹⁰. This suggests that the high normal stresses applied in the LS regime may be responsible for the observed non-Newtonian behavior of the confined aqueous solution.”

Moreover, the paragraph regarding the thermo-activated slip regime, on lines 130-138, is not written in a way that is accessible to a wide audience.

Part of this is stylistic and could be addressed by improving the writing to make more fluid and idiomatic, but a second part of this involves the heavy use of jargon and very specialized language.

Please, explain the “transition state theory”.

What is a net frequency?

Adjacent energy minimum?

Thank you very much. We have modified the description and have added a cartoon to explain this mechanism. It reads now:

“IS regime: Shear-promoted thermally-activated slip

FIGs. 1f-j show the friction force as a function of velocity in the IS-regime ($\sigma_n \sim 200-400$ MPa). The friction force scales with the logarithm of velocity, i.e. $F_L \sim \log(v)$, thereby indicating a pathway of energy dissipation that differs from the viscous dissipation in the LS regime. This logarithmic relation is attributed to a shear-promoted thermally-activated slip in the context of transition state theory²⁰. This theory was originally developed by Eyring²¹ to describe the reaction rate at the molecular level and is often applied to describe the origin of friction in both dry contacts¹⁴ and thin-film lubrication²². That is, for slip to occur, the molecule, initially in an equilibrium position (an energy minimum) needs to jump over a “transition state” before reaching the adjacent energetic minimum, which requires an activation energy (ΔE) (FIG. 2a). Although the thermal energy of the molecules might be sufficient to overcome this energy barrier, the shear force applied on the molecule reduces ΔE , and thereby, promotes the slip; when the molecule falls in the adjacent minimum, the applied work is irreversibly dissipated (lost). Although described at the molecular level, this theory has been often applied to explain friction in macroscale contacts^{23,24}.

Based on our previous work¹⁶, we assume that hydrated calcium ions are not squeezed-out under high applied pressures, but instead they remain localized on oppositely charged surface sites and undergo similar thermally-activated slip, as shown in FIG. 2a. Considering that the slip rate of the molecules is increased by the applied lateral force according to $v \sim v_0 \exp(-(\Delta E - F_L \lambda)/k_B T)$, the following expression¹⁰ is obtained for the friction force F_L :

$$F_L = \frac{\Delta E}{\lambda} + \frac{k_B T}{\lambda} \log(v/v_0) \quad \text{Eq. 2}$$

k_B being the Boltzmann constant, T the absolute temperature, v_0 the molecular fluctuation rate at a reference state, and λ the stress-activation length. λ represents the characteristic length over which the confined ions and water are capable of building up strain in response to shear, and it appears in FIG. 2a as the elongation from the initial equilibrium position to the transition state. The solid lines in FIGs. 1. f-j show the fits of Eq. 2 to the experimental results. The calculated λ significantly decreases with increasing normal loads at all investigated concentrations (FIG. 2b); that is, the localized ions and water can build up strain over shorter distances, which suggests a decrease in mobility of the confined solution with increase in normal stress.

FIG. 2. a) Schematics of the shear-promoted thermally-activated slip in the IS-regime ($\sigma_n \sim 200-400$ MPa). b) Stress-activation length λ obtained from fitting Eq. 2 to the velocity-dependent friction force in the IS-regime. The region under the grey dashed line shows the range of λ in dry experiments, which are also described by the shear-promoted thermally activated slip of the surface atoms. Tip radius=150 nm. c) Crossover velocities v_c in the HS regime ($\sigma_n > 400$ MPa); tip radius=100 nm. The results in b-c) are shown for the selected CaCl_2 concentrations. “

6) L326-380: The discussion on the implications of the authors’ results to fault lubrication is probably the weakest point of the manuscript. First, I do not really understand the message that the authors want to convey. I believe they tried to compare nano-scale friction to cm-scale friction experiments. However, I have the impression that the relationship between these 2 measurements is not obvious. For instance, the friction coefficients measured in this study are ~ 0.01 whereas it is assumed that calcite bearing rocks have a friction coefficient around 0.6. Can the authors comment on this point?

Thank you for your comments. We agree that this part needed improvement.

We propose that pressure solution of calcite provides a weakening mechanism of the fault strength based on our friction-force measurements on a single-asperity contact. This simplified system allows us decoupling the mechanism of pressure solution from other processes that concur in cm-scale friction measurements. We agree with the reviewer that the relationship between these two types of measurements is not obvious. The cm-scale friction force results from multiple processes occurring at the same time such as compaction, plastic deformation, granular flow, intergranular lubrication, and asperity interlocking. We only evaluate a single asperity contact in the absence of compaction, plastic deformation and granular flow, which yields much smaller coefficients of friction. We have modified the manuscript as the reviewer suggested and we hope you find that the main message we try to convey is now delivered. The discussion has been modified accordingly.

Minor comments:

L 9: shear loading

Thank you very much. We have revise “undergoing shear” with “under shear loading”

L10: remove : “understanding.... relevance”.

Thanks. We have deleted this sentence.

L22: fault irreversible deformation

Thanks. Following the indication of the editor we have simplified the introduction to make it more accessible to readers and this sentence has been modified.

“When the static friction between two adjacent tectonic plates under shear loading is overcome, the two plates move relative to each other, which is known as a fault slip; under conditions of unstable slip, that is, when an intermittent motion (or *stick-slip*) occurs, an earthquake rupture may take place¹.”

L26: Rate and State Friction (RSF)
Thank you. We have corrected this.

L27: RSF is used to understand earthquake nucleation, not propagation.
Thank you. We have changed the “earthquake phenomena” into “earthquake nucleation”.

L28: granular material = gouge
Thank you. We have changed “granular materials” into “gouges”.

L33-36: “aging of silica-silica... across the interface”. Please avoid the use of jargon and very specialized language.

Thank you. We have modified the sentence as:

“For instance, a previous AFM study demonstrated that the origin for the increase in the static friction of single-asperity silica-silica contacts of nanometer size is the formation of chemical bonds between the silica surfaces²⁵⁻²⁷.”

L 62: “located in an acoustic chamber”. Please avoid the use of jargon and very specialized language. Explain better the experimental protocol.

Thank you. To simplify the description, we deleted “located in an acoustic chamber”. We have added a section with the improved method to the manuscript.

L93: I do not understand what ‘confinement geometry’ means.

By “a confinement geometry”, we mean a limited space between two confining surfaces. But we appreciate that the reviewer pointed this out, and this sentence has been deleted from the manuscript. We say now: “By sliding an oxidized silicon tip along the atomically flat calcite (10 $\bar{1}$ 4) plane, the friction force was measured while isolated from other processes occurring concurrently in natural systems and macroscale experiments.”

Fig1: Figure 1 and figure 4 should be merged to a single figure that must show the 3 deformation regimes
Fig S9 should be within the main text and inserted before figure 1. This should help the reader understanding why the authors are talking about water lubrication.

We appreciate the reviewer’s suggestion very much. As you will find in the revised version of the manuscript, we have merged Fig. 1 and 4 into a new Figure 1 with all the stress regimes. The dry experiment is now shown in Figure 4, as also suggested.

L142-146: 3 different definitions of the coherence length are given. Please simplify and give only one definition.

Yes. You are right:

“..., and λ the stress-activation length. λ represents the characteristic length over which the confined ions and water are capable of building up strain in response to shear, and it appears in FIG. 2a as the elongation from the initial equilibrium position to the transition state”.

FIG. 1 and FIG. 4: curves 10nN and 20nN are plotted in both fig1 and fig4. Please precise if 10nN and 20 nN are in the IL or in the HL regime.

Thank you very much. We would like to clarify that the pressure dictates the change of regime (and not the load). To clarify this, we have changed the names of the regimes to “low stress”, “intermediate stress” and “high stress” regimes (LS, IS and HS). In AFM, the maximum load is limited by the linearity of the

stiffness of the cantilever, and hence, we need to limit the range of applied loads. Therefore, the measurements in the HL regime were conducted with a tip of smaller radius to be able to apply higher pressures (at the same load). We agree that this was confusing in the manuscript.

We explain now:

“FIGs. 1k-o show the friction force as a function of velocity in the HS-regime ($\sigma_n > 400$ MPa). Note that, owing to the smaller radius of the tip used in these experiments (100 nm, instead of 150 nm in IS-regime), the contact stress is higher than in the IS-regime ($\sigma_n \sim 200$ -400 MPa) at the same applied load.”

Fig.5: please plot the dry experiments on fig. 5.

Thank you for this suggestion. We have included the dry friction in Fig. 5a.

Fig.5: In calcite friction coefficient is assumed to be ~ 0.5 . Can the author explain the low ratio between the friction force and the friction load (~ 0.01)?

As now explained:

“Previous studies of calcite-rich fault gouge reported a decrease in the friction coefficient from ~ 0.7 to ~ 0.6 , i.e. much less pronounced¹¹. This is, however, not surprising, since granular flow, asperity interlocking and fracture as well as plastic deformation of multiple contacts can concurrently happen in both dry and wet experiments with (simulated) fault gouges and rock surfaces under shear loading, and they significantly increase friction¹². The simple geometry of our study enables us to exclude these complex phenomena, and hence, to evaluate the direct effect of hydration lubrication and pressure solution on the frictional characteristics of calcite.”

Fig.5: please explain better the panels b, c, d

Thank you. We believe that the dedicated explanation for friction loops in the Methods section will facilitate the understanding. We have also expanded the discussion of panels d, c, and d as the following:

“In summary, we have resolved three different mechanisms responsible for the frictional strength of this nanoscale asperity. FIG. 5a shows the friction force divided by load as a function of the slip rate in the three regimes; note that in the HS-regime this value significantly differs from the friction coefficient, which becomes negative under certain conditions (FIG. 3a). Representative measurements for the lateral force as a function of the sliding distance (during trace and retrace of the tip) at three selected sliding velocities are shown in FIGs. 5b-d. Hydration lubrication occurs at normal stresses $\sigma_n \leq 200$ MPa, when the confined solution lubricates the nanoscale contact and friction is small and dictated by the viscous shear force of the fluid film. Here, the corrugation of the lateral force (FIG. 5b-d, blue) is indistinguishable from the noise (~ 0.02 nN), supporting smooth sliding, and thus, viscous energy dissipation. At higher normal stresses ($\sigma_n \sim 200$ -400 MPa, green), velocity-strengthening friction is originated by the shear-promoted thermally-activated slip of the surface-localized hydrated ions, and it is strongly dependent on the solution composition. Here, the variation of the lateral force is much higher than the noise, revealing a periodic stick-slip with a slip length of ~ 0.5 nm throughout all sliding velocities. Such discontinuous sliding is characteristic of the shear-promoted thermally-activated slip in nanoscale experiments as a result of the slip of the molecules from the transition state into the energy minimum^{28,29}. Intermittent slip prevails in the HS-regime (red), although it becomes more irregular, which is likely associated to the concurrent pressure solution of calcite. The pressure-induced dissolution re-activates hydration lubrication at higher stresses ($\sigma_n > 400$ MPa), thereby causing a significant weakening of the frictional strength of this single asperity. Here, the dashed line shows the calculated friction force assuming that the friction force at the highest velocities is still given by Eq. 2 and extrapolating to lower velocities; the area between calculated and measured friction force gives the decrease in dissipated energy per unit time as a result of pressure solution. (see arrows in FIG. 5a). This is most prominent at slow sliding velocities ($0.2 \mu\text{m/s}$, FIG. 5b), where the lateral force in the HS-regime is of similar magnitude as in the LS- and IS-regimes but at much higher contact stresses. The lateral force increases when sliding occurs at $1 \mu\text{m/s}$, while the reproducible drops pointed by the arrows in FIG. 5c reflect the sequential weakening of the frictional strength upon

sliding. The characteristic drops vanish with a further increase in the sliding velocity to 5 $\mu\text{m/s}$ (FIG. 5d), indicating that pressure-solution facilitated slip is not noticeably active anymore. This mechanism is more pronounced with increasing calcium concentration, likely owing to the enhanced calcite surface reactivity; in nature, it will depend on the specific composition of the brine.”

L 307-310: I am not really convinced that Fig. 5b shows discontinuous sliding. I see only noise. The noise of our AFM is ~ 0.02 nN, which is similar to the corrugation of the lateral forces in the LS-regime. Note that this noise does not depend on the applied load and velocity. This noise is much lower than the amplitude observed in IL- and HL-regimes, and therefore, we conclude that Fig. 5b shows discontinuous sliding. We note that this discontinuity is characteristic of thermally activated slip in nanoscale experiments as a result of the sudden jumps of the molecules from the transition state into the adjacent energetic minima^{28,29}.

- 1 Brace, W. F. & Byerlee, J. D. Stick-Slip as a Mechanism for Earthquakes. *Science* **153**, 990-&, doi:DOI 10.1126/science.153.3739.990 (1966).
- 2 Ruina, A. Slip Instability and State Variable Friction Laws. *Journal of Geophysical Research* **88**, 359-370, doi:DOI 10.1029/JB088iB12p10359 (1983).
- 3 Marone, C. Laboratory-derived friction laws and their application to seismic faulting. *Annu Rev Earth Pl Sc* **26**, 643-696, doi:DOI 10.1146/annurev.earth.26.1.643 (1998).
- 4 Pina, C. M., Miranda, R. & Gnecco, E. Anisotropic surface coupling while sliding on dolomite and calcite crystals. *Physical Review B* **85**, 073402, doi:ARTN 073402 10.1103/PhysRevB.85.073402 (2012).
- 5 Audet, P. & Schwartz, S. Y. Hydrologic control of forearc strength and seismicity in the Costa Rican subduction zone. *Nature Geoscience* **6**, 852-855 (2013).
- 6 Scholz, C. H. *The Mechanics of Earthquakes and Faulting*. 2 edn, (Cambridge University Press, 2002).
- 7 Scuderi, M. M., Collettini, C. & Marone, C. Frictional stability and earthquake triggering during fluid pressure stimulation of an experimental fault. *Earth and Planetary Science Letters* **477**, 84-96, doi:<https://doi.org/10.1016/j.epsl.2017.08.009> (2017).
- 8 Royne, A., Bisschop, J. & Dysthe, D. K. Experimental investigation of surface energy and subcritical crack growth in calcite. *J Geophys Res-Sol Ea* **116**, doi:Artn B04204 10.1029/2010jb008033 (2011).
- 9 Li, T. D. & Riedo, E. Nonlinear viscoelastic dynamics of nanoconfined wetting liquids. *Phys Rev Lett* **100**, 106102, doi:10.1103/PhysRevLett.100.106102 (2008).
- 10 Ma, L., Gaisinskaya-Kipnis, A., Kampf, N. & Klein, J. Origins of hydration lubrication. *Nature Communications* **6**, 6060 (2015).
- 11 Verberne, B. A. *et al.* Frictional Properties and Microstructure of Calcite-Rich Fault Gouges Sheared at Sub-Seismic Sliding Velocities. *Pure and Applied Geophysics* **171**, 2617-2640, doi:10.1007/s00024-013-0760-0 (2014).
- 12 Ruths, M., Berman, A. D. & Israelachvili, J. N. in *Nanotribology and Nanomechanics: An Introduction* (ed Bharat Bhushan) 389-481 (Springer Berlin Heidelberg, 2005).
- 13 Yang, C.-Y. Calcium and Magnesium in Drinking Water and Risk of Death From Cerebrovascular Disease. *Stroke* **29**, 411 (1998).
- 14 Hardie, L. A. Origin of CaCl₂ Brines by Basalt-Seawater Interaction - Insights Provided by Some Simple Mass Balance Calculations. *Contributions to Mineralogy and Petrology* **82**, 205-213, doi:Doi 10.1007/Bf01166615 (1983).
- 15 Chhabra, R. P. in *Rheology of complex fluids* 3-34 (Springer, 2010).
- 16 Diao, Y. J. & Espinosa-Marzal, R. M. Molecular insight into the nanoconfined calcite-solution interface. *Proceedings of the National Academy of Sciences of the United States of America* **113**, 12047-12052, doi:10.1073/pnas.1605920113 (2016).
- 17 Israelachvili, J. N. & Pashley, R. M. Molecular Layering of Water at Surfaces and Origin of Repulsive Hydration Forces. *Nature* **306**, 249-250 (1983).

- 18 Alcantar, N., Israelachvili, J. & Boles, J. Forces and ionic transport between mica surfaces: Implications for pressure solution. *Geochimica Et Cosmochimica Acta* **67**, 1289-1304, doi:Doi 10.1016/S0016-7037(02)01270-X (2003).
- 19 Røyne, A., Dalby, K. N. & Hassenkam, T. Repulsive hydration forces between calcite surfaces and their effect on the brittle strength of calcite-bearing rocks. *Geophysical Research Letters* **42**, 4786-4794 (2015).
- 20 Eyring, H. The Activated Complex in Chemical Reactions. *The Journal of Chemical Physics* **3**, 107-115, doi:10.1063/1.1749604 (1935).
- 21 Eyring, H. Viscosity, plasticity, and diffusion as examples of absolute reaction rates. *The Journal of chemical physics* **4**, 283-291 (1936).
- 22 Briscoe, B. & Evans, D. in *Proc. R. Soc. Lond. A.* 389-407 (The Royal Society).
- 23 Spikes, H. & Tysoe, W. On the Commonality Between Theoretical Models for Fluid and Solid Friction, Wear and Tribochemistry. *Tribology Letters* **59**, 21, doi:10.1007/s11249-015-0544-z (2015).
- 24 Schallamach, A. Recent Advances in Knowledge of Rubber Friction and Tire Wear. *Rubber Chemistry and Technology* **41**, 209-244, doi:10.5254/1.3539171 (1968).
- 25 Li, Q., Tullis, T. E., Goldsby, D. & Carpick, R. W. Frictional ageing from interfacial bonding and the origins of rate and state friction. *Nature* **480**, 233-236 (2011).
- 26 Tian, K. W. *et al.* Load and Time Dependence of Interfacial Chemical Bond-Induced Friction at the Nanoscale. *Physical Review Letters* **118**, 076103, doi:ARTN 076103 10.1103/PhysRevLett.118.076103 (2017).
- 27 Liu, Y. & Szlufarska, I. Chemical origins of frictional aging. *Phys Rev Lett* **109**, 186102, doi:10.1103/PhysRevLett.109.186102 (2012).
- 28 Gnecco, E. *et al.* Velocity dependence of atomic friction. *Phys Rev Lett* **84**, 1172-1175, doi:10.1103/PhysRevLett.84.1172 (2000).
- 29 He, M., Blum, A. S., Overney, G. & Overney, R. M. Effect of interfacial liquid structuring on the coherence length in nanolubrication. *Phys Rev Lett* **88**, 154302, doi:10.1103/PhysRevLett.88.154302 (2002).

REVIEWERS' COMMENTS:

Reviewer #4 (Remarks to the Author):

This is a substantially revised manuscript. The authors have done a very nice job of improving the science communicated in the paper. Fig1, Fig3 and Fig5 are much clearer. I think the paper can be published as it is, after the authors have a chance to consider the following minor comments:

- 1) L25 Cite Scholz, *The Mechanics of Earthquakes and Faulting*, 2002
- 2) L 37: Cite Violay et al. 2013 EPSL; Violay et al., 2015 EPSL
- 3) L41: Cite Brodsky & Kanamori, JGR 2001 ; Bayart et al., 2016 Physical review letter
- 4) L41-42-, "Velocity-weakening friction and stick-slip were observed in experiments on water saturated limestone gouge at high temperature (150°C), whereas stable sliding was observed under dry and low temperature conditions"
- 5) L70/L163): brine concentration
- 6) L289, "FIG. 5a shows the friction force divided by load as a function of the slip rate in the three regimes; dry friction is given for comparison"
- 7) Fig5 b: explain in the figure caption the color code (red, green blue curves)
- 8) L 336-339: it is not clear to me how weakening mechanisms (granular flow, fracture, plastic deformations, subcritical crack growth,) can strengthen the fault interfaces. Please explain better why the friction coefficient observed in AFM experiments (0.01) is much lower than the friction coefficient measured on the field or in classical rock mechanics tests (0.6-0.8).
- 9) L359: how is it possible to be in velocity stegthening regime and observed at the same time stick slip events (L298)?
- 10) I do not understand the term "disjoining pressure"

Dear reviewer,

We are thrilled to know that this manuscript merits the publication in Nature Communications. Thank you very much for all the comments so far, which have significantly improved our work.. We have addressed each comment point by point in this document and have tracked the changes in the manuscript.

Sincerely,

Rosa Espinosa-Marzal

Reviewer 4's comments:

This is a substantially revised manuscript. The authors have done a very nice job of improving the science communicated in the paper. Fig1, Fig3 and Fig5 are much clearer. I think the paper can be published as it is, after the authors have a chance to consider the following minor comments.

1) L25 Cite Scholz, The Mechanics of Earthquakes and Faulting, 2002

We have added this reference.

2) L 37: Cite Violay et al. 2013 EPSL; Violay et al., 2015 EPSL

We sincerely thank the reviewer for providing these references. We have added the first reference. However, the second reference is not in the range of slip rate relevant to earthquake nucleation. Therefore, we decide not to include it to avoid any confusion.

3) L41: Cite Brodsky & Kanamori, JGR 2001; Bayart et al., 2016 Physical review letter

We have added these references.

4) L41-42-, "Velocity-weakening friction and stick-slip were observed in experiments on water saturated limestone gouge at high temperature (150°C), whereas stable sliding was observed under dry and low temperature conditions"

We have modified this sentence accordingly.

5) L70/L163): brine concentration

We have modified these two phrases accordingly.

6) L289, "FIG. 5a shows the friction force divided by load as a function of the slip rate in the three regimes; dry friction is given for comparison"

We have modified this sentence accordingly.

7) Fig5 b: explain in the figure caption the color code (red, green blue curves)

We have added an explanation to the caption.

8) L 336-339: it is not clear to me how weakening mechanisms (granular flow, fracture, plastic deformations, subcritical crack growth,) can strengthen the fault interfaces. Please explain better why the friction coefficient observed in AFM experiments (0.01) is much lower than the friction coefficient measured on the field or in classical rock mechanics tests (0.6-0.8).

Thank you for pointing this out. The discrepancy between nanoscopic and macroscopic friction coefficients has been reported for other substrates in an aqueous environment previously. In macroscopic sliding experiments, a coefficient experiment of 0.23 (Horn et al., 1962) was measured between mica surfaces, while Ma et al. (2015) reported a friction coefficient as low as 0.0001 in SFA measurements. A coefficient of friction as high as 0.6 was reported for quartz gouge (Chester et al., 1994) while colloidal AFM friction measurements reported $\mu \sim 0.1$ between silica surfaces (Donose et al., 2006). Another study aimed to compare macro- and nanoscale frictional behavior of graphite; the corresponding friction coefficients were 0.2 and 0.01, respectively (Liu et al. 1998). The origin for the discrepancies in the coefficient of friction has

been proposed to rely on surface topography and roughness, plastic deformation, wear, and material transfer, that are not or do not occur at the same extent at the two scales.

We have modified the paragraph as follows:

“We note the significantly lower friction coefficients reported in our work compared to macroscale friction coefficients in fault gouge and rocks. There are multiple reasons for such discrepancy, as demonstrated previously for other substrates like mica^{1,2}, quartz^{3,4}, and graphite⁵. In the case of fault gouge and rock friction, asperity interlocking during macroscopic sliding of such rough surfaces and asperity fracture can cause a significant increase in friction that is excluded in atomically smooth contacts, like in our nanoscale experiments. The simple geometry of our study enables us to evaluate the direct effect of hydration lubrication and pressure solution on the frictional characteristics of calcite at the level of a single asperity.”

9) L359: how is it possible to be in velocity strengthening regime and observed at the same time stick slip events (L298)?

We report here an atomic-scale stick-slip with a slip length of ~0.5 nm. Such discontinuous sliding is characteristic of the shear-promoted thermally-activated slip in nanoscale experiments when measured by AFM, and it is the result of the jumps of the tip from the transition state to the adjacent energy minimum, and hence, it is the characteristic of a velocity-strengthening frictional behavior.⁶ We explain this in the manuscript.

10) I do not understand the term “disjoining pressure”

The disjoining pressure is an excess pressure in the confined fluid film (excess compared to the bulk) due to the colloidal forces acting between the confining surfaces (like electrical double layer, hydration force etc.).

We have defined the term “disjoining pressure” in L97 as follows:

“This work showed that a strong repulsion between the confining surfaces –also termed disjoining pressure – is originated by the colloidal forces between the two confining surfaces across the thin film of solution, and it prevents the solution to be squeezed-out under an applied pressure⁷⁻⁹.”

References

- 1 Ma, L., Gaisinskaya-Kipnis, A., Kampf, N. & Klein, J. Origins of hydration lubrication. *Nat. Commun.* **6**, 6060 (2015).
- 2 Horn, H. M. & Deere, D. U. Frictional Characteristics of Minerals. *Géotechnique* **12**, 319-335, doi:10.1680/geot.1962.12.4.319 (1962).
- 3 Chester, F. M. Effects of temperature on friction: Constitutive equations and experiments with quartz gouge. *J Geophys Res: Sol Ea* **99**, 7247-7261, doi:10.1029/93JB03110 (1994).
- 4 Donose, B. C., Vakarelski, I. U., Taran, E., Shinto, H. & Higashitani, K. Specific effects of divalent cation nitrates on the nanotribology of silica surfaces. *Ind. Eng. Chem. Res.* **45**, 7035-7041 (2006).

- 5 Liu, E., Blanpain, B., Celis, J.-P. & Roos, J. Comparative study between macrotribology and nanotribology. *J. Appl. Phys.* **84**, 4859-4865 (1998).
- 6 Izabela, S., Michael, C. & Robert, W. C. Recent advances in single-asperity nanotribology. *Journal of Physics D: Applied Physics* **41**, 123001 (2008).
- 7 Israelachvili, J. N. & Pashley, R. M. Molecular layering of water at surfaces and origin of repulsive hydration forces. *Nature* **306**, 249-250 (1983).
- 8 Alcantar, N., Israelachvili, J. & Boles, J. Forces and ionic transport between mica surfaces: Implications for pressure solution. *Geochim. Cosmochim. Acta* **67**, 1289-1304, doi:Doi 10.1016/S0016-7037(02)01270-X (2003).
- 9 Røyne, A., Dalby, K. N. & Hassenkam, T. Repulsive hydration forces between calcite surfaces and their effect on the brittle strength of calcite-bearing rocks. *Geophys. Res. Lett.* **42**, 4786-4794 (2015).